# The methylome of the model arbuscular mycorrhizal fungus, *Rhizophagus irregularis*, shares characteristics with early diverging fungi and Dikarya

Anurag Chaturvedi [1,4,5], Joaquim Cruz Corella[1,5], Chanz Robbins[1,5], Anita Loha[2], Laure Menin[3], Natalia Gasilova[3], Frédéric G. Masclaux[1], Soon-Jae Lee [1] & Ian R. Sanders [1✉]

Early-diverging fungi (EDF) are distinct from Dikarya and other eukaryotes, exhibiting high N6-methyldeoxyadenine (6mA) contents, rather than 5-methylcytosine (5mC). As plants transitioned to land the EDF sub-phylum, arbuscular mycorrhizal fungi (AMF; Glomeromycotina) evolved a symbiotic lifestyle with 80% of plant species worldwide. Here we show that these fungi exhibit 5mC and 6mA methylation characteristics that jointly set them apart from other fungi. The model AMF, *R. irregularis*, evolved very high levels of 5mC and greatly reduced levels of 6mA. However, unlike the Dikarya, 6mA in AMF occurs at symmetrical ApT motifs in genes and is associated with their transcription. 6mA is heterogeneously distributed among nuclei in these coenocytic fungi suggesting functional differences among nuclei. While far fewer genes are regulated by 6mA in the AMF genome than in EDF, most strikingly, 6mA methylation has been specifically retained in genes implicated in components of phosphate regulation; the quintessential hallmark defining this globally important symbiosis.

[1] Department of Ecology and Evolution, University of Lausanne, Lausanne, Switzerland. [2] Department of Plant Molecular Biology, University of Lausanne, Lausanne, Switzerland. [3] Institute of Chemical Sciences and Engineering, Swiss Federal Institute of Technology Lausanne (EPFL), SSMI, Batochime, Lausanne, Switzerland. [4] Present address: Environmental Genomics Group, School of Biosciences, The University of Birmingham, Birmingham, UK. [5] These authors contributed equally: Anurag Chaturvedi, Joaquim Cruz Corella, Chanz Robbins. ✉email: ian.sanders@unil.ch

The early diverging fungi (EDF), comprising several phyla, are ancient and thought to be up to one billion years old[1,2]. Their methylation patterns are mostly distinct from the two later diverging fungal phyla making up the Dikarya, as well as most other eukaryotes, as they exhibit high levels of N6-methyldeoxyadenine (6mA) and very low levels of 5-methylcytosine (5mC)[1]. Thus, this major genomic divergence between these two groups during the evolution of the fungal kingdom is also mirrored in their epigenome. Like in bacteria, 6mA in EDF is primarily located at symmetrical ApT motifs in gene bodies; a feature that is associated with active gene transcription and also likely allows inheritance of 6mA epigenetic marks during DNA replication[1]. The Dikarya have lost these features of 6mA methylation and, like in many other Eukaryotes, 5mC is considered to play a more important role in gene regulation. The Glomeromycotina are considered as a clade of the EDF[3]. Known as arbuscular mycorrhizal fungi (AMF), what sets them apart from other fungal groups is that they have been entirely symbiotic with terrestrial plants ever since plants colonized land ~450 million years ago and are thought to have played a role in this major evolutionary transition[4]. The most-comprehensive studies of fungal 6mA and 5mC methylation have detailed the prevalence of these epigenetic marks across the fungal phylogeny, including both EDF and Dikarya[1,5]. Despite being one of the most important fungal clades, no Glomeromycotina species were included in those studies (Fig. 1). While phylogenetic studies clearly place AMF in the Mucoromycota clade of EDF[6], they share some lifestyle and genomic features with the Dikarya. First, they form dikaryons[7]. Second, a homeodomain-containing mating locus resembles those found in the Basidiomycota, rather than in the Mucoromycota[7,8]. AMF are major drivers of plant diversity and global carbon and nutrient cycles[9–11]. Numerous studies have shown the capacity of AMF to increase plant P and N acquisition as well as growth[12,13]. Inoculation of globally important crops with AMF increases overall yields[14], while reducing the necessity for P fertilization[15]. Given the rapidly depleting global stocks of P[16,17], this highlights their importance for global food security. Although studies have revealed important characteristics of AMF genomes, little is known to date on their methylome nor are there any insights into its functional implications.

The model AMF species *Rhizophagus irregularis* has been the focus of genome sequencing studies[18]. Large intra-specific genome differences exist within this fungus[7,19–22]. This variation is considered biologically important as it leads to significant differences in fungal quantitative growth traits, P uptake and transfer to plants, and plant productivity[15,19,23,24]. However, some *R. irregularis* isolates are coenocytic homokaryons, harboring multiple identical haploid nuclei, whereas others are coenocytic dikaryons, harboring a population of two haploid nucleus genotypes[7,8,25]. Clonal offspring of *R. irregularis* dikaryons have been shown to enormously alter the growth of rice by up to fivefold and cassava by up to threefold[15,26]. Such variation could be due to the inheritance of different proportions of the two nucleus genotypes[26,27]. However, clonal offspring of homokaryon *R. irregularis*, which are genetically identical, also induce equally large differences in cassava growth; pointing strongly to the role of epigenetic variation in symbiotic effects of these fungi[15].

Here, we used the Pacific Biosciences RSII platform with SMRT cell technology to compare structural variation (SV) and 6mA epigenomic variation among genetically different *R. irregularis* isolates, between two clones, and between two nuclei genotypes within a dikaryon isolate. The epigenomic characteristics across the fungal kingdom have previously comprised only one isolate of a species as representative for each of the main fungal clades, probably owing to the high costs of resolving 6mA epigenomes

with this platform. However, because of the biological interest in elucidating 6mA methylation differences among isolates, among clones, and among nuclei within the fungi, this study requires a particularly large number of SMRT cells. We performed whole-genome sequencing on six *R. irregularis* isolates (A1, A5, B12, C2, C5, and C3). We present a study on the 6mA methylome in AMF, characterizing these epigenetic marks in *R. irregularis* isolates C2, C5, and C3. These isolates show strongly differential effects on plant growth[15], even though C2 and C5 are genetically indistinguishable at the SNP level[20]. Isolates C3 and A5 are dikaryons[7,25,27]. Because these fungi have haploid nuclei[7], single-molecule sequencing revealing variation within a dikaryon represents variation between the two nucleus genotypes.

## Results and discussion

**Genome coverage, annotation, and SV**. Details of genome coverage, annotation, assembly, and SV are given in: Supplementary Notes 1 and 2, Supplementary Figs. S1–S5, Supplementary Tables S1–S6, and Supplementary Data 1. Analysis of SV confirmed the previously published phylogenies and very high relatedness between isolates C2 and C5[20,22] (Supplementary Fig. S1; Supplementary Note 3). Analysis of SV also allowed us to show that C2 and C5 are genetically indistinguishable and were, therefore, considered as clones (Supplementary Fig. S4). It also allowed us to define the structural differences between nuclei in the dikaryon C3 (Supplementary Note 4). SV between nucleus genotypes shared the same characteristics as seen between isolates and gene SV was as large as that seen, between some isolates (Supplementary Fig. S5a-d).

**The amounts of 6mA and 5mC methylation in AMF are unlike the other EDF**. Having confirmed that C2 and C5 were likely clones, and defined the structurally different regions of the two nucleus genotypes in C3, we characterized 6mA methylation. Sufficient coverage (25× per strand) allowed reliable detection of 6mA in C2, C5, and C3 using Pacific Biosciences SMRT Analysis software kineticsTools (Supplementary Fig. S2).

Approximately 0.2% of adenine was methylated in *R. irregularis* (Fig. 2a). Liquid chromatography–mass spectrometry (LC-MS) confirmed similar proportions of 6mA methylation among isolates (0.12 to 0.17%) (Fig. 2a). Although the Glomeromycotina are considered EDF, 6mA methylation in *R. irregularis* was much lower than most of the rest of this ancient fungal group and was in the range typically seen in Dikarya, plants, and animals[1,28–31] (Fig. 2a). A high abundance of 5mC methylation is expected when 6mA levels are low[1,5] and, indeed, we observed 32.5–49.5% of cytosine was methylated. This characteristic was consistent with the Dikarya and other Eukaryotes, in the sense that it is high, but not with the other EDF[1] (Fig. 2b). However, the percentage of 5mC exceeds that recorded in other Dikarya to date[5]. There are two species in the EDF where similarly low 6mA levels have been observed (*Rhizoclosmatium globosum* and *Catenaria anguillula*), but unlike *R. irregularis* both show very low levels of 5mC methylation[1]. A higher prevalence of 5mC methylation than in other fungal species suggests an important role of this epigenetic mark in gene regulation in AMF and, thus, requires further characterization.

**Among-nucleus 6mA methylation heterogeneity is a conserved feature**. While 6mA-methylated sites in most other EDF show full methylation, 6mA methylation in *R. irregularis* was highly heterogeneous, with methylation ratios from 0.64 in C3 to 0.68 in C5 (Fig. 2c). Coverage had a negligible influence on the ratio (Supplementary Fig. S6). Some 6mA heterogeneity has been reported in one member of the EDF (*Lobosporanguim transversal*; Mortierellomycotina), but not to the amount observed in *R.*

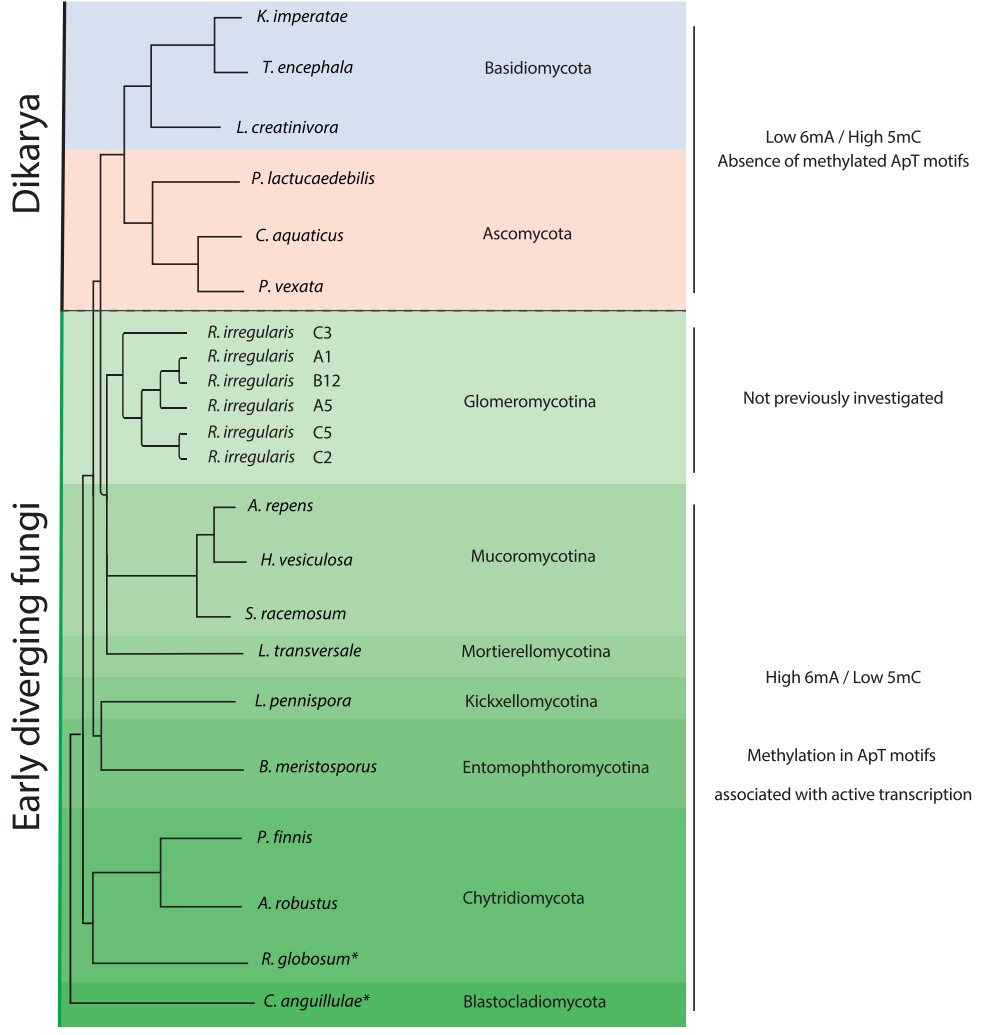

**Fig. 1 Previously investigated 6mA and 5mC contents throughout the fungal phylogeny.** A phylogeny of the fungal kingdom showing that there was a shift in the functional methylome during the divergence of EDF and Dikarya from high 6mA and low 5mC contents in EDF to to high 5mC and low 6mA contents in Dikarya. However, the arbuscular mycorrhizal fungi (Glomeromycotina) have not been previously investigated. *Represents species that do not follow the general pattern of methylation in EDF. In these species, 6mA is low and appears not to have the same functional significance as in EDF as it does not occur in ApT motifs, which are important for ensuring active gene transcription.

*irregularis*. Pairwise comparisons of methylation ratios between isolates showed that only a negligible proportion of methylated sites exhibited a significantly diverging methylation ratio, highlighting the robustness of these methylation heterogeneity estimates. Only 0.62%, 1.41%, and 1.29% of methylated sites revealed diverging methylation ratios between C2 and C5, between C2 and C3, and between C5 and C3, respectively (at 2.5% and 97.5% confidence limits). Conducting single-molecule sequencing means that each sequenced DNA strand originates from one haploid nucleus. The haploid status of nuclei in *R. irregularis* means that the observed epigenetic heterogeneity in AMF must be partitioned among the population of nuclei, thus giving a variable function among nuclei within one coenocytic fungus. 6mA methylation heterogeneity among cells in prokaryotes allows population-level adaptation to arise from clonal colonies[32,33], and in eukaryotes, among-cell methylation heterogeneity is crucial for differentiation[34,35]. Thus, if 6mA has a role in gene regulation, the pattern of among-nucleus 6mA heterogeneity represents a layer of variation in *R. irregularis* that could explain the extremely plastic response in the symbiotic effects of clonal sibling lines of these fungi on plant growth and allow a plastic response of the fungus to the environment.

**6mA methylation occurs in a core fungal gene set but methylation differences also occur among clones and also between nuclei in dikaryon isolates**. Because both genetically different and identical *R. irregularis* isolates show differential effects on plant growth, we investigated their 6mA methylation differences. Specifically, 22.8%, 22.8%, and 21.9% of genes in C2, C5, and C3, respectively, harbored 6mA. Out of a 290 BUSCO core fungal gene set, 270 of those genes occurring in *R. irregularis* were methylated. A total of 7062 genes were methylated in the three isolates (Fig. 2d). All genes methylated in at least one isolate were genes occurring in all isolates. Thus, 6mA methylation acts on a core gene set, similar to other EDF, rather than isolate-specific genes. Most of these (4398 genes; 62%) were methylated in all isolates (Fig. 2d). However, 31% were differentially methylated between C3 and C2/C5. The methylation repertoire was not identical between clones C2 and C5 (Fig. 2d), meaning that epigenetic variation could cause differential gene regulation between genetically identical isolates. SV analysis between two clones will never show two individuals to be completely identical because of sequencing errors, coverage, or problems with the reference assemblies. Although the very small number of SVs were discounted as artefacts (Supplementary Note 3), we checked

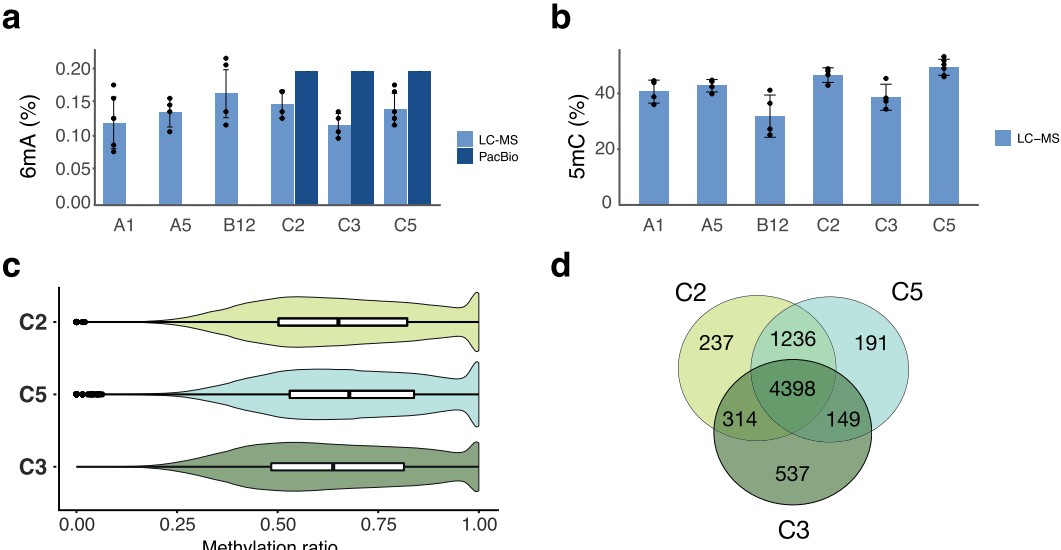

**Fig. 2 Quantification of 6mA and 5mC in *R. irregularis* isolates. a** Percentage of total methylated adenine (6mA) was detected by single-molecule DNA sequencing in three isolates (C2, C5, and C3) and validated by LC–MS in six isolates. The data are available in Supplementary data 3. **b** Percent of total methylated cytosine (5mC) detected by LC–MS in six isolates. The data are available in Supplementary data 3. **c** Violin plot (with boxplot inside) showing the distribution of the methylation ratio of *R. irregularis* isolate C2 ($n = 291654$; minima:0, maxima:1, mean:0.6594, Q1:0.5020, Q3:0.8220), C5 ($n = 240110$; minima:0, maxima:1, mean:0.6813, Q1:0.5300, Q3:0.8390) and C3 ($n = 263696$; minima:0, maxima:1, mean:0.6465, Q1:0.4840, Q3:0.8140) suggesting that 6mA methylation in *R. irregularis* was highly heterogeneous. **d** Venn diagram representing numbers of common and unique genes harboring 6mA ApT motifs in isolates C2, C5, and C3. The majority of methylated genes were shared among isolates, yet considerable differences were observed among isolates and between clones C2 and C5.

to see if differential methylation in those SVs could result in the 6mA methylation differences observed between C2 and C5. Methylation only occurred in 22 SVs and only one of these contained a gene body. Thus, differential methylation in arte-factual SVs cannot explain the methylation differences between these two isolates. If 6mA regulates gene expression, then this could at least partially explain the extremely large differences in symbiotic effects of clones C2 and C5 on crop growth, as well as among clonal siblings[15]. Differential methylation also occurred between nuclei within the dikaryon C3 (Supplementary Note 5).

**6mA methylation in AMF has a likely functional role in gene regulation similar to other EDF.** Although 6mA and 5mC abundances set AMF apart from most other EDF, several char-acteristics point to a similar role of 6mA in gene regulation in *R. irregularis* as in other EDF rather than Dikarya. First, 6mA marks in *R. irregularis* occurred as symmetrical methylation at ApT di-nucleotide motifs (73% in C3; 68% in C2 and C5), where AATT was the most abundant (Fig. 3a; Supplementary Data 2). This prevalent epigenomic 6mA signature across the EDF and unicellular algae are distinct from the Dikarya and animals[1,28,29]. Furthermore, just like in other EDF, and unlike plants and animals[36,37], 6mA sites at these ApT motifs were strongly biased to positions within genes and in the upstream promoter region of the genes (~86% of 6mA-ApT sites)(Fig. 3b). These two non-random patterns of 6mA distribution strongly suggest a functional significance as observed in other EDF. Although two species in other EDF clades (*Rhizoclosmatium globosum* and *Catenaria anguillulae*) also have low 6mA abundance, they either have low or no, ApT motif methylation[1]. That ApT methylation is symmetrical means that methylation can be retained on par-ental strands during DNA replication and nucleus division and is, thus, likely heritable; a feature that was lost in the later diverging Dikarya.

The non-random patterns of 6mA distribution strongly suggest a likely functional consequence of 6mA symmetrical ApT signatures

in the Glomeromycotina epigenome. Indeed, 6mA symmetrical ApT motif signatures (with a mean value of 10 ApT di-nucleotide motifs per gene; Supplementary Fig. S7) in genes were associated with transcription of those genes. Most genes containing 6mA ApT motifs were highly expressed in isolates C2 and C3, irrespective of 6mA abundance (total amount of 6mA) in the gene (fragments per kilobase million; FPKM > 5) (Fig. 3c), while a significant positive correlation occurred between gene 6mA methylation density (methylation abundance divided by gene length) and transcription levels (Kendall's correlation τ: 0.3649, $P < 0.001$ and τ: 0.4395, $P < 0.001$ in C2 and C3, respectively).

**Methyltransferases in AMF are more similar to EDF than Dikarya.** Mondo et al.[1] studied the differences in counts between EDF and Dikarya for pfam domains that are thought responsible for fungal 6mA methylation. They found that 44 pfam domains significantly differed ($p < 0.01$). Among them, the MT-A70 family (PF05063) has a central role as a DNA 6mA methyltransferase[31]. A subclade of the MT-A70 protein family, namely AMT1, is responsible for genome-wide DNA methylation levels and the maintenance of symmetric 6mA methylation in ApT motifs in eukaryotes[38]. Interestingly, Wang et al.[38] showed that two EDF species had an AMT1 homolog, whereas one species of Dikarya did not. Because symmetric methylation in ApT motifs is a distinct feature of 6mA methylation in EDF compared with the Dikarya, we performed a phylogenetic analysis of the MT-A70 family across the EDF and the Dikarya with a larger set of fungal taxa compared with Wang et al.[38] including all the fungal species reported in Mondo et al.[1], previously classified reference sequences from Wang et al.[38] and six additional Glomer-omycotina species (including the six isolates of *R. irregularis*). The previously characterized and classified AMTs protein sequences from Wang et al.[38] were used as internal controls to identify the different MT-A70 subclasses.

As expected, we found that EDF, including AMF, had a larger set of MT-A70 family proteins compared with Dikarya. Moreover,

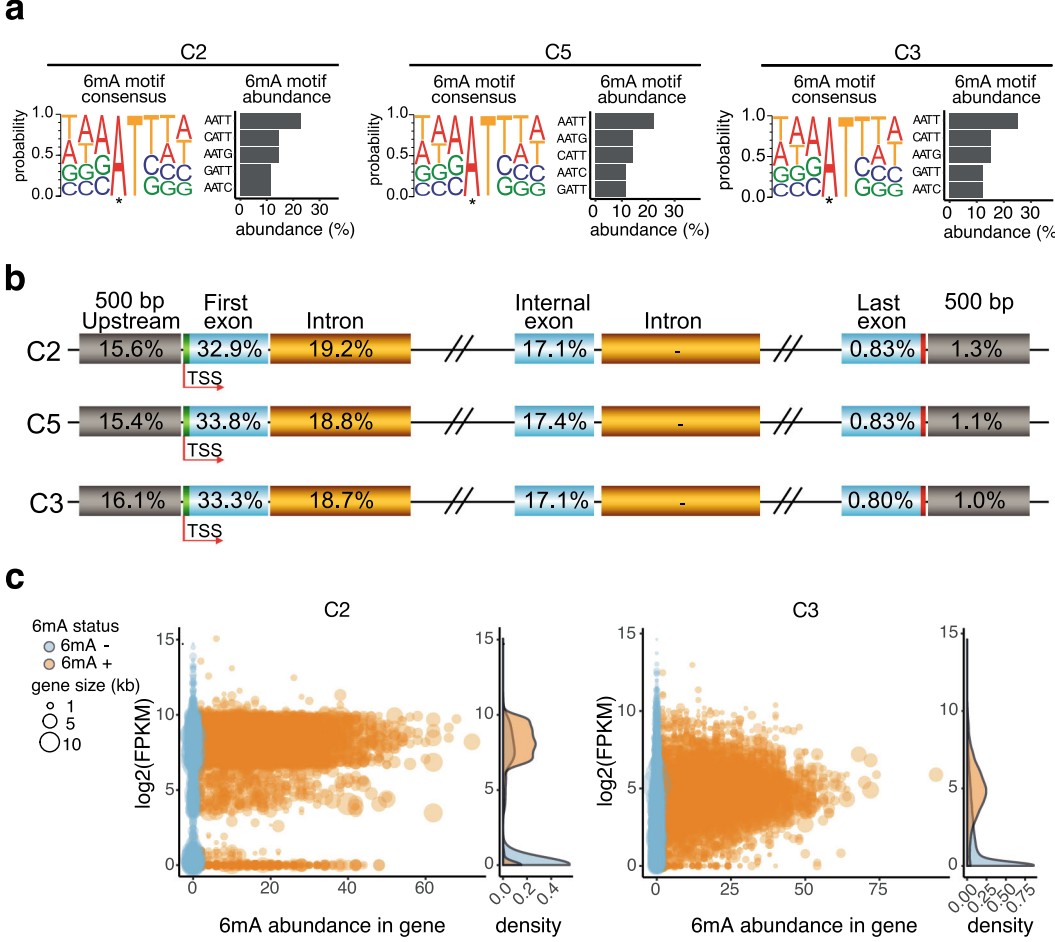

**Fig. 3 Characterization of 6mA in *R. irregularis* isolates. a** 6mA consensus sequence motifs with the probability of occurrence of each motif and the most abundant motifs in isolates C2, C5, and C3 arranged in decreasing abundance. Several ApT motifs other than AATT, CATT, AATG, GATT, and AATC were also present in each of the three isolates and in similar proportions among the three isolates. * Represents the methylated base. Data available in Supplementary Data 2. **b** Topological distribution of 6mA ApT dinucleotides in the gene structure in C2, C5, and C3. **c** Gene expression (log₂(FPKM)) in isolates C2 and C3 of genes based on 6mA ApT abundance within gene bodies (6mA + : orange; 6mA-: blue). Gene size is represented by circle size and density curves representing the occurrence of genes with 6mA abundance (6mA + : orange; 6mA−: blue) with respect to gene expression log₂(FPKM) values. The data are available in Supplementary Data 4.

AMT1 was found in all EDF and AMF that exhibit substantial symmetric ApT methylation, whereas no homolog was found in Dikarya (Fig. 4). Two EDF, *C. anguillulae* and *R. globosum*, exhibit no, or extremely low, 6mA methylation in ApT motifs, thus, exhibiting a pattern more similar to Dikarya. We show that *R. globosum* harbors a distinctively short AMT1 homolog (Supplementary Fig. S8), and no AMT1 homolog was found in *C. anguillulae*, which is likely why this species has almost no symmetric 6mA methylation.

All AMF genomes we analyzed contained a conserved AMT1 homolog as in other EDF, which could explain the high degree of symmetric methylation at ApT motifs despite the fact that 6mA levels are low in *R. irregularis*. The abundance of 6mA methylation and 5mC methylation of fungal genomes is inversely correlated[1], suggesting a relationship between 6mA MTases and 5mC MTases in fungi, with a higher rate of losing than gaining 5mC MTases when an AMT1 homolog is present[5]. *R. irregularis* appears to be an exception where conservation of a highly symmetric ApT methylation is coupled with a high degree of 5mC methylation. The AMF clade has all five conserved 5mC MTase types (DNMT1, DNMT2, DNMT5, DIM-2, and RID), whereas sister clades of the Mucoromycota phylum have lost DNMT5 and RID[5]. Together with the divergent 6mA

methyltransferases repertoire reported in our study, the conservation of methyltransferases in *Rhizophagus irregularis* is in line with the high degree of 5mC methylation found in this fungus and low 6mA level.

**6mA gene methylation is biased towards specific gene functions in AMF, including the regulation of DNA methylation.** Patterns of among-clone, and within-individual, 6mA heterogeneity could explain why genetically identical AMF clones and siblings of AMF homokaryons display enormous differences in their effects on crop growth. However, although Glomeromycotina 6mA methylation appears to have a similar regulatory role as in other EDF, it affects a much smaller number of genes than in other EDF with the notable exception of *Linderina pennispora*. Thus, it was important to investigate whether 6mA methylation affects genes known to affect the functioning of the symbiosis. GO analysis of 6mA harboring genes showed enrichment of transferase activity, binding, catalytic activity, DNA methyltransferase activity, hydrolase activity (including pyrophosphatase activity), and transporter activity (Fig. 5a; Supplementary Data 1). We also found enrichment of both ion and ATPase transmembrane transporter activity (Figs. 5b and 5c), which is significant given the beneficial effect of AMF are

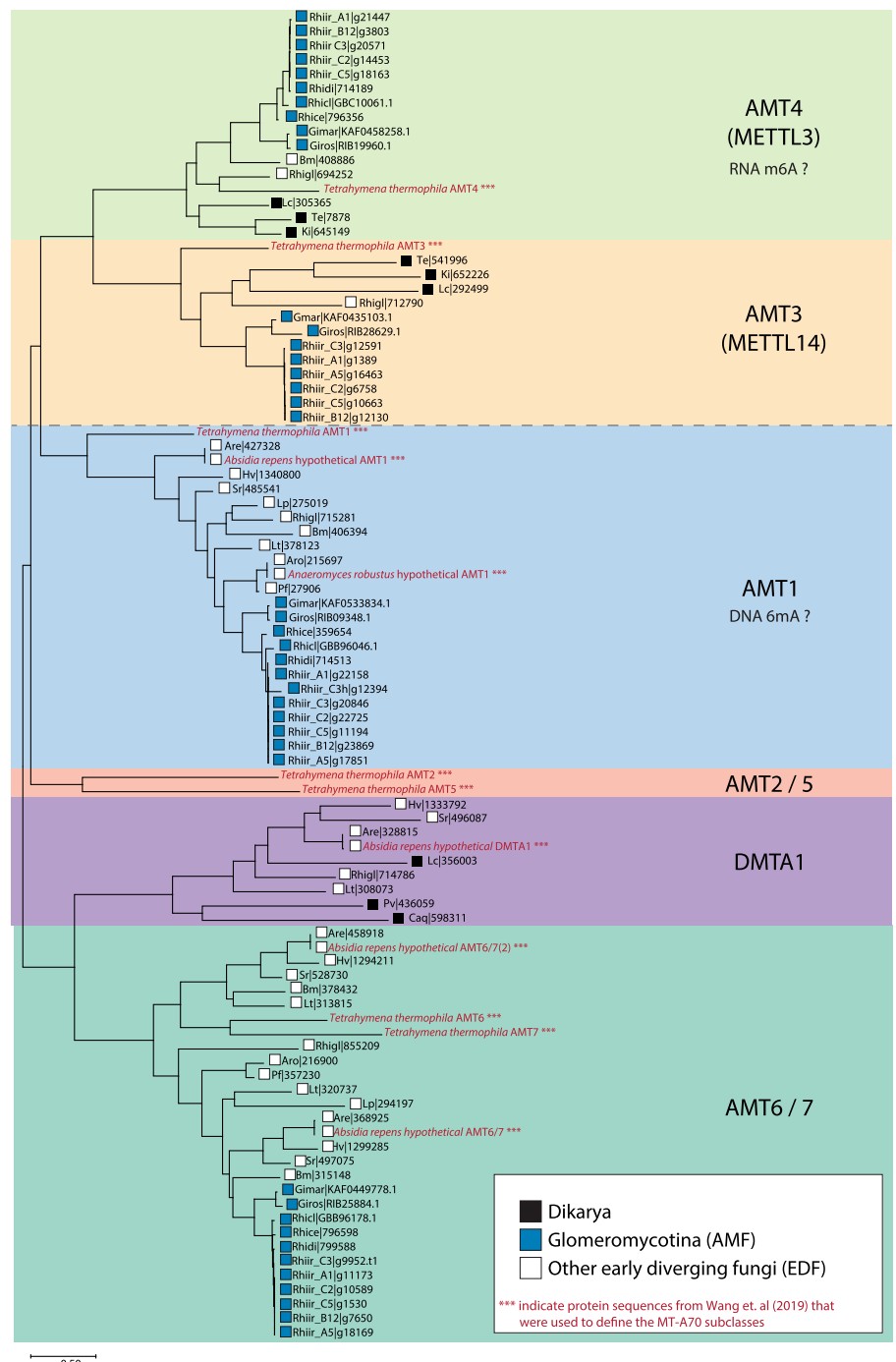

**Fig. 4 Phylogenetic analysis of MT-A70 proteins across fungal taxa.** Putative RNA m6A MTases (METTL3, METTL14) and putative DNA 6mA MTase subclasses (AMT2/5, AMT1, DMTA1, and AMT6/7) are separated by a dashed line. Protein sequences of *T. thermophila*, *A. repens*, and *A. robustus* from Wang et. al.[38] were used to define the subclasses of MT-A70 proteins. These sequences are colored in red and marked with three asterisks. Dikarya, early diverging fungi (EDF), and AMF taxa are specified with black, white, and blue squares, respectively. Species abbreviations are as follows: *Clohesyomyces aquaticus* (Caq); *Leucosporidiella creatinivora* (Lc); *Protomyces lactucaedebilis*: (Pl); *Pseudomassariella vexata* (Pv); *Tremella encephala* (Te); *Kockovaella imperatae* (Ki); *Catenaria anguillulae* (Can); *Hesseltinella vesiculosa* (Hv); *Linderina pennispora* (Lp); *Lobosporangium transversale* (Lt); *Piromyces finnis* (Pf); *Syncephalastrum racemosum* (Sr); *Absidia repens* (Are); *Anaeromyces robustus* (Aro); *Basidiobolus meristosporus* (Bm); *Rhizoclosmatium globosum* (Rhigl); *Rhizophagus cerebiforme* (Rhice); *Rhizophagus clarus* (Rhicl); *Rhizophagus diaphanus* (Rhidi); *Rhizophagus irregularis* isolate A1 (Rhiir_A1); *Rhizophagus irregularis* isolate A5 (Rhiir_A5); *Rhizophagus irregularis* isolate B12 (Rhiir_B12); *Rhizophagus irregularis* isolate C2 (Rhiir_C2); *Rhizophagus irregularis* isolate C3 (Rhiir_C3); *Rhizophagus irregularis* isolate C3 (secondary haplotig, pseudo-secondary nucleus genotype) (Rhiir_C3h); *Rhizophagus irregularis* isolate C5 (Rhiir_C5); *Gigaspora margarita* (Gimar); *Gigaspora rosea* (Giros).

based on the fungus ability to absorb nutrients and transport them to the plant. In addition, we observed the 6mA methylation of genes enriched for DNA methyltransferase, suggesting that the transfer of methyl groups to DNA (potentially 5mC, as well as 6mA) could itself be under the control of 6mA methylation.

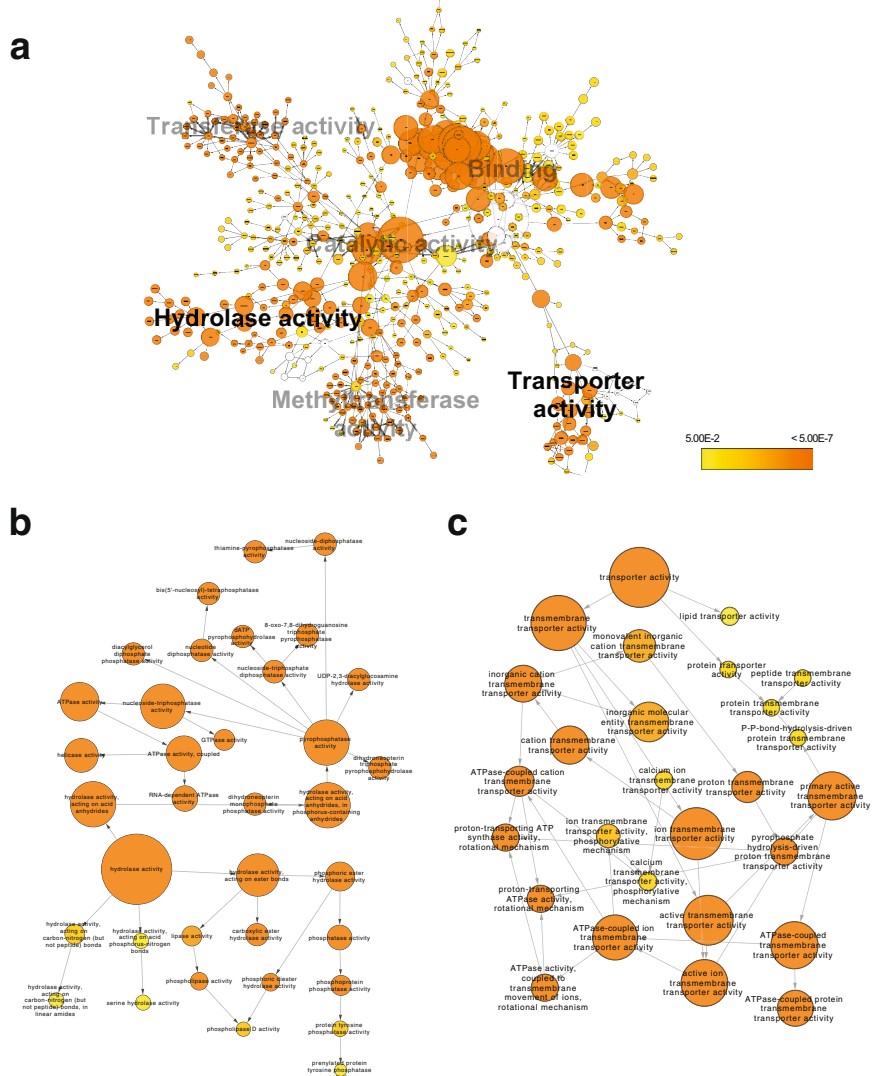

**Fig. 5 Network of enriched Gene ontology (GO) molecular function categories of commonly methylated 6mA ApT genes in *R. irregularis* isolates C2/C5 and C3. a** General overview of the six principal enriched clusters of molecular function with the hydrolase activity and transporter activity clusters highlighted in bold. **b** Hydrolase activity cluster shown in greater detail. **c** Transporter activity cluster shown in greater detail. The diameter of the circle refers to a number of genes per GO term (which is shown in Supplementary Data 1) and the color of the circle refers to the FDR corrected probability (*P*), where the yellow to orange gradient represents increasing significance (Supplementary Data 1).

**AMF have specifically retained 6mA methylation in genes important for the mycorrhizal symbiosis.** The quintessential hallmark of mycorrhizal symbiosis is that the fungi absorb and transport P to the plant. RNAi and transcription studies have revealed Glomeromycotina genes involved in P transport, metabolism, sensing and signaling, homeostasis, and acquisition pathways[39,40]. Here, we observed that some of these genes contained SV, but the vast majority appear under the regulation of 6mA methylation (Supplementary Table S7; Supplementary Note 6). These included low-affinity SPX domain-containing P transporters (PHO87, PHO88, PHO90, PHO91). SPX domains are considered key to P transport and metabolism in eukaryotes including AMF[39,41]. Remarkably, all the genes identified in previous studies as being involved in inositol polyphosphate synthesis/hydrolysis and P responsive signaling appear to be under epigenetic control. Such genes are thought to be extremely important in AMF because these fungi have to take up and regulate P levels much greater than those needed by the fungus alone[39]. After P absorption by the fungus, AMF has to convert phosphate into polyphosphate. The vacuolar transporter chaperones VTC1, VTC

2, and VTC4, involved in polyphosphate synthesis (and also containing SPX domains), are also under 6mA epigenetic control. Finally, the parts of PKA signaling pathways, that are strongly upregulated in AMF in response to low P environments, and the MAPK and Tor signaling pathways, that are downregulated in low P environments appear to be almost completely methylated, likely under epigenetic control on the regulation of these pathways in the fungus in varying P environments. The fact that *R. irregularis* has specifically retained 6mA methylation in the genes regulating the key resource, phosphorus, that the plant needs from this symbiosis, while clearly having lost 6mA methylation from a very large number of other genes is highly intriguing. Partners in mutualistic symbioses are expected to evolve mechanisms to prevent over-exploitation by the partner. Given the high 5mC levels in *R. irregularis*, a large part of gene regulation in AMF must be under 5mC control rather than 6mA. Plants have evolved mechanisms to regulate gene expression through 5mC and use mobile small RNA (sRNA) molecules to regulate 5mC methylation[42]. Cross-kingdom transfer of mobile sRNA molecules is a commonly recognized way that organisms in plant–microbe interactions manipulate gene

expression of the partner[43]. Indeed, a recent study of AMF genomes revealed that AMF possesses an RNAi system that regulates sRNA, as in plants[44]. We suggest that if such sRNA transfer occurs between plants and AMF, which is highly likely, then by retaining a 6mA regulatory mechanism to regulate access to the key resource needed by plants, AMF may prevent host manipulation of the genes controlling this resource and allow the fungus to determine the amount of P available to plants.

## Conclusions

As Glomeromycotina diverged from other EDF ~500 million years ago and formed symbioses with the first land plants, they evolved epigenomic features akin to the Dikarya, such as high 5mC abundance and low levels of 6mA methylation. However, the combination of 5mC levels higher than those recorded in Dikarya, and low 6mA levels, coupled with 6mA methylation characteristics not seen in other EDF that harbor low 6mA, make the methylome patterns in *R. irregularis* unique in the fungi. Despite the loss of 6mA from a huge number of genes, the Glomeromycotina retained heritable EDF-like 6mA gene regulation capabilities in a conserved core set of genes, including those fundamental to their globally important role in symbiosis and P cycling as well as in those genes controlling methylation of DNA; possibly even 5mC, which is the most abundant part of the Glomeromycotina methylome. Methylated adenine in Glomeromycotina is not randomly distributed, occurring symmetrically near genes, and is associated with gene transcription. Despite the conservation of 6mA features in the Glomeromycotina, a significant amount of heterogeneity in 6mA methylation among nuclei exists, as well qualitatively differential 6mA among genes in genetically identical clones. These features point to a clear mechanism to explain the enormous differences that genetically indistinguishable AMF and clonal sibling fungi induce in the production of globally important crops.

## Methods

**DNA isolation, library preparation, and sequencing**. A mixture of fungal hyphae and spores from each of the *R. irregularis* isolates was used for all DNA extractions. The biological material was produced in vitro and either provided by Symbiom (Lanškroun, Czech Republic) (A5, B12, C2, C3, C5) or produced in-house using a split-plate culture system (A1)[45]. The material of each of the isolates was collected at the same time and stored at 4°C in sterile ddH$_2$O until use.

In a laminar flow hood, we took 1 mL of a spore suspension, and as much water was removed as possible, before being flash frozen. Each sample was pulverized using cryogenic grinding (precooling: 2 min. 5 Hz; grinding1: 30 sec. 25 Hz; intermission: 30 sec. 5 Hz; grinding 2: 30 sec. 25 Hz) with a Retsch CryoMill (9.739 299, Retsch GmbH, Germany) and were stored in liquid Nitrogen in preparation for DNA extraction.

DNA extraction was performed using the MagAttract® HMW DNA Mini Kit (67563 Qiagen) in exactly the same way for each isolate. In brief, the protocol was scaled up to account for starting material of ~100 mg. After MB buffer was added, the protocol was modified to increase incubation time at 25°C to 10 min. at 800 rpm. After the removal of the first supernatant, volumes were used as written in the protocol, and only incubation times were increased to 2 min. at 800 rpm. The last two steps of the protocol were performed according to the manual and cut tips were used for transferring high molecular weight DNA. Individual extractions were then pooled for each individual and purified once with Agencourt AMPure XP® (Beckman Coulter) magnetic beads using 1× volumes of beads. DNA was then quantified with Quantus™ Flurometer (Promega) and analyzed for fragment length using Fragment Analyzer (Advanced Analytical Technologies, Inc.).

High molecular weight DNAs were sheared in a Covaris g-TUBE (Covaris, Woburn, MA, USA) to obtain ~20 kb fragments. After shearing the DNA size distribution was checked on a Fragment Analyzer (Advanced Analytical Technologies, Ames, IA, USA). In all, 5 μg of the sheared DNA was used to prepare a SMRTbell library with the PacBio SMRTbell Template Prep Kit 1 (Pacific Biosciences, Menlo Park, CA, USA) according to the manufacturer's recommendations. The resulting libraries were size selected on a BluePippin system (Sage Science, Inc. Beverly, MA, USA) for molecules larger than 15 kb. The resulting libraries were sequenced with P4/C2 chemistry and MagBeads on a PacBio RSII system (Pacific Biosciences, Menlo Park, CA, USA) at 240 min movie-length using SMRT cells v2.

**Genome assemblies and annotation**. The genome assemblies of homokaryon isolates (A1, B12, C2, C5) and dikaryon isolates (A5 and C3) were conducted using two separate bioinformatics software, i.e., HGAP4 and FALCON-Unzip, respectively. Specifically, HGAP4-based assemblies of A1, B12, C2, and C5[46] were scaffolded using SSPACE[47] followed by polishing with PILON[48] using Illumina reads from[21]. BlobTools[49] was used for the identification and removal of non-fungal sequence contaminations. The resulting scaffolds were renamed in descending order based on their length. Sequence data of the dikaryon isolates A5 and C3 was first assembled using FALCON-Unzip[50], which is a diploid-aware genome assembler that generates two genome assemblies consisting of one primary genome and a secondary haplotig. This software uses SMRT long-read data to identify heterozygous regions in the genome and constructs haplotigs. Thereafter, the resulting assemblies were similarly scaffolded as homokaryons using SSPACE and polished using PILON. After removal of non-fungal sequence contaminations using BlobTools, scaffolds were sorted and renamed according to their length. Augustus[51] was trained with publicly available RNA sequencing data of C2, C3, and DAOM197198[25] and used for gene predictions on all assemblies. The functional GO term categories were identified using reciprocal BLAST[52] searches for genes in the published gene annotations of AMF isolates[21]. The resulting assemblies and annotations are accessible through the European Nucleotide Archive under the accession number: PRJEB33553.

The phylogeny of six *R. irregularis* isolates was constructed as follows. We first identified conserved gene families with one gene per isolate using OrthoFinder[53] ($n = 6941$). Each single-copy gene from each family was then aligned with MAFFT software using default settings[54]. Further cleaning and concatenation of alignments were performed using GBLOCKS v0.91b[55]. RAxML[56] was then used to calculate phylogenetic distances using the PROTGAMMAWAG protein model and trees were constructed by bootstrapping with 1000 iterations to provide node support and visualized in dendroscope[57] and geneious[58].

**Mapping of raw reads and variant calling for structural variant detection**. To detect and report the SV (defined in this study as >30 bp) among the six isolates of *R. irregularis* (A1, A5, B12, C2, C5, and C3) we choose to compare SV to a reference isolate DAOM197198[21]. Although the threshold length for detection of an SV is arbitrary, we used a threshold of >30 bp as this is in line with contemporary studies using long-range sequence data[59,60]. We used CoNvex Gap-cost alignMents for Long Reads (NGMLR) to align the raw reads on the reference genome, and Sniffles to call the SV from each of the alignment files[61]. We ran both tools with default parameters and filtered for translocations and all variants that had low read support (RE < 15).

In addition, higher sequencing depth in isolate C3 allowed the analysis of SV within a dikaryon isolate of AMF. In order to identify these SVs, we first mapped raw sequencing data of C3 on the primary assembly of C3 using NGMLR. Variants were then called by Sniffles and the same filtering described above was applied to discard translocations and variants with low read support.

**Characterization of SV in *R. irregularis***. First, we counted the number of SVs that were reported in the callset of each isolate for the four following classes: insertions (INS), deletions (DEL), inversions (INV), and duplications (DUP).

Second, we computed the Jaccard distances among the isolates by performing all possible pairwise comparisons of their SV callsets. We accounted for the number of SVs with a reciprocal overlap of at least 80% (BEDtools)[62] and of the same SV class. Hierarchical clustering was then performed based on the Jaccard distances with the R package "pvclust"[63], by bootstrapping with 10,000 iterations.

Third, in order to find out which structural variants were related to the presence of repeated sequences in the genome, we used the available repeat annotation of isolate DAOM197198 using the BEDTools suite. TE-related SVs were determined by the presence of their breakpoints within the span of an annotated TE in isolate DAOM197198. Finally, gene annotation of isolate DAOM197198 was used to determine the genes intersecting SVs.

In the case of the SV within the dikaryon isolate C3, we computed the same metrics described above. Variants related to repeated sequences were determined on the basis of repeat annotation of C3 primary assembly. Similarly, genes directly affected by SVs were detected on the basis of the gene annotation model of C3.

**Comparison of SVs between isolates C2 and C5**. Because previous studies had documented the high genetic relatedness of the two *R. irregularis* isolates C2 and C5, we investigated more in detail whether or not SVs existed between them to best determine whether they were actually clones. Two different in silico approaches were used to investigate possible structural differences between the genomes of these two isolates.

First, we compared the two final SV callsets of each isolate generated after comparison with the reference isolate DAOM197198. We counted the number of SVs that were co-occurring in the same locations in both isolates. Then, we manually curated all the SVs that initially appeared to be isolate specific using IGV.

Second, in order to reduce the chance of observing false positives, we repeated the mapping of the raw sequence data of C2 and C5 using NGMLR to the newly assembled genome of C2. Then, SV was called using Sniffles with default parameters.

Finally, we manually curated all the SVs that appeared to be isolate specific using IGV.

**Epigenetic modification detection and data analysis.** The Pacific biosciences RSII platform reads were converted from native bax format to subread bam format using bax2bam SMRT Analysis software. Next, pbalign software was used for mapping subreads to genome assemblies of C2, C5 and C3 using algorithm options to be "--bestn 10 --minMatch 12 --maxMatch 30 --minSubreadLength 50 --minAlnLength 50 --minPctSimilarity 70 --minPctAccuracy 75 --hitPolicy randombest --concordant --randomSeed 1 --useQuality --minPctIdentity 70.0". Tools analyzing polymerase kinetics were then used for the detection of 6mA signatures with minimum sequencing 25× per strand coverage with $p$ value of 0.001. All the tools can be accessed at: https://github.com/PacificBiosciences.

All detected 6mA signatures were analyzed for the degree of variation (methylated AT content divided by total AT content), topology (localization of ApT motifs within the gene body), and functional annotation (based on GO analysis). Counting the number of genes affected by methylation, we only considered genes to be methylated which harbor ApT di-nucleotide motif (with a mean value of 10 ApT di-nucleotide motifs per gene; Supplementary Fig. S7). The BEDTools suite was used for the characterization of 6mA topology along the gene body. The gene body includes the start site of the gene to the end site of the gene with a 500 bp upstream (promoter region) and a 500 bp downstream region as used in Mondo et. al[1]. There is a lack of complete information regarding gene boundaries that includes regulatory regions in *R. irregularis*, thus, the limit that represents the gene body has to be set arbitrarily.

To test whether 6mA presence was positively correlated with gene transcription in *R. irregularis*, we analyzed transcriptome data generated from isolates C2 and C3[25] and the growth conditions for both RNA-seq and DNAseq (long-read sequencing) were exactly the same. RNA sequencing data were mapped on the genome assemblies of each isolate using Tophat[64]. Cufflinks[65] was used for transcript assembly and transcript abundance analysis, using default parameters and isolate-specific gene models, respectively. Statistical analysis was performed using R version 3.5.3. Kendall's correlation coefficient τ and statistical significance were calculated between 6mA density in the gene (number of sites of 6mA/length of the coding region for the gene) and gene expression (FPKM) by package "Kendall". Statistical significance was assumed at the 95% level.

**LC-MS/MS analysis.** Approximately 1 μg genomic DNA was sequentially digested to render individual nucleosides. After incubation at 95°C for 3 min., DNA was cooled on ice for 1 min. and digested overnight at 42°C (1 U DNase I from bovine pancreas with included buffer; Roche Diagnostics, Indianapolis, USA). Following digestion, 3.4 μL of 1 M NH$_4$HCO$_3$ and 1 U phosphodiesterase (from *Crotalus adamanteus* venom and reconstituted in 110 mM Tris HCl (pH 8.9), 110 mM NaCl, 15 mM MgCl$_2$; Sigma-Aldrich) were added to break phosphodiester bonds and incubated at 37°C for 2 h. Immediately after, 2 U alkaline phosphatase (from bovine intestinal mucosa; Sigma-Aldrich) was added followed by another round of incubation at 37°C for 2 h. Fully digested nucleosides were passed through 0.22 μm cellulose acetate filter (Costar® Corning Inc., Salt Lake City, USA), to remove enzymes. After the first round of filtration, the internal standard N$^{15}$-labeled form of m4dC was added to samples and then filtered one final time using Ultrafree-MC GV Durapore (Merck Millipore, Cork, Ireland).

Quantitative analysis of global levels of dC, m5dC, dA, and m6dA was performed on a high-resolution Q Exactive HF Orbitrap FT-MS instrument (Thermo Fisher Scientific, Bremen, Germany) coupled to a Dionex UltiMate 3000 UPLC system (Thermo Fisher Scientific, Bremen, Germany). Analytes were separated on an Acquity UHPLC HSS T3 column (100 × 2.1 mm, 1.8-μm particle size). The mobile phase consisted of 0.1% aqueous formic acid (solvent A) and 0.1% formic acid in acetonitrile (solvent B) at a flow rate of 300 μl/min. Calibration curves (ranging from 0.1 nM to 500 nM) were generated with serial dilutions of synthetic standards of target compounds. The mass spectrometer was set in positive-ion mode and operated in parallel reaction monitoring. Ions of masses 228.10 (dC), 242.11 (m5dC), 252.11 (dA) and 266.12 (m6dA) were isolated with 2 m/z isolation window and fragmented by HDC (higher-energy collisional dissociation) with NCE (normalized collision energy) of 28%. Full MS/MS scans were acquired with resolution 30000 for the base fragments 112.0508, 126.0664, 136.0617, and 150.0774 m/z (cytosine, methylcytosine, adenine, and methyladenine, respectively) with 5 p.p.m. mass. The extracted ion chromatogram of the base fragment was used for quantitation. The accurate mass of the corresponding base fragment was extracted with the XCalibur Qual Browser and XCalibur Quan Browser software (Thermo Scientific) and used for quantification. In addition, peak area integration was manually curated and corrected when necessary. All measurements were performed in duplicate and quantities were represented as mean values of two replicates. To estimate the percentage of methylated cytosine and methylated adenine, we divided the value of methylated cytosine and methylated adenine, quantified in nM by the total amount of cytosine and adenine (methylated and non-methylated) present in the sample, respectively.

**Identification of putative DNA 6mA methyltransferases in *R. irregularis*.** We investigated all MT-A70 family hits (PF05063, pfam version 33.1) in 22 fungal species

from Mondo et al.[1] and 11 AMF isolates and species (six isolates from the present study and five AMF species for which genomes were available from the joint genome institute (JGI) and Genbank. The sources of five AMF species from JGI and Genbank are the following: *Rhizophagus cerebiforme* DAOM 227022 (https://mycocosm.jgi.doe.gov/Rhice1_1), *Rhizophagus diaphanous* (https://mycocosm.jgi.doe.gov/Rhidi1), *Rhizophagus clarus* (GCA_003203555.1), *Gigaspora margarita* (GCA_009809945.1), *Gigaspora rosea* (GCA_003550325.1). The protein family search was conducted by hmmsearch, from HMMR v3.0[66], with default parameters and retained only significant hits ($e$ value <0.05, for the full sequence; $e$ value < 0.01, for the best 1st domain). Then, we retrieved the protein sequences from their corresponding fungal catalog, and performed a multiple sequence alignment using MUSCLE[67] with default parameters. Protein sequences of, *A. repens* and *A. robustus* AMTs (MT-A70 MTases) described by Wang et. al.[38] were also included in the alignment and *T. thermophila* AMTs were used for internal references to define each sub-class of AMTs. The phylogenetic analysis of these proteins was performed using MEGA X[68], where we used the sequence alignment file to construct a maximum likelihood tree, using the default parameters.

**Gene ontology enrichment and analysis of SV and 6mA in candidate genes.** We performed gene ontology (GO) enrichment analysis to find out whether there was an over-representation of given biological processes (BP) and molecular functions (MF) among the genes that presented SV within their sequences and those that contained 6mA epigenetic marks. Published DAOM197198 GO annotations were used for GO enrichment of genes containing SV. Published GO annotations of isolate C2 (representing both isolates C2 and C5 because they are genetically indistinguishable) and published isolate A4 GO annotations (representing isolate C3 because A4 and C3 are genetically indistinguishable) were used as a background for GO analysis of genes containing 6mA epigenetic marks. The BINGO plugin of Cytoscape [69,70] was used for this purpose. The GO terms with $P \leq 0.05$, after Benjamini & Hochberg (FDR) correction, were considered to be over-represented BP and MF categories. In addition, we investigated individual genes for the presence/absence of SV and 6mA epigenetic modification in a set of P and sugar transporters and other genes that are involved in P acquisition and metabolism. The annotation of these genes was based on molecular functions in GO categories and sequenced-based blast analysis using AMF genes[39,40,71].

**Statistics and reproducibility.** For LC-MS/MS-based quantification of 6mA and 5mC, at least 1 μg of genomic DNA was used for sample preparation for all six *R. irregularis* isolates (A1, A5, B12, C2, C5, and C3). Quantification of 6mA and 5mC was performed in triplicate or duplicate and quantities were represented as mean values of the replicates. The percentage estimation of methylated cytosine and methylated adenine was performed by dividing the value of methylated cytosine and methylated adenine, quantified in nM by the total amount of cytosine and adenine (methylated and non-methylated) present in the sample, respectively.

The Kendall's correlation between gene 6mA methylation density (methylation abundance divided by gene length) and transcription levels was performed using R software while maintaining a significance of $P < 0.001$.

**Reporting summary.** Further information on research design is available in the Nature Research Reporting Summary linked to this article.

## Data availability
The raw data, resulting in genome assemblies, and annotations are accessible through the NCBI BioProject under the accession number: PRJEB33553. Source data underlying figures are presented in Supplementary Data 1–4.

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

## Acknowledgements
This work was supported by Swiss National Science Foundation grants (project numbers 31003A_162549/1 and 310030B_182826/1) whose support we gratefully acknowledge.

## Author contributions
A.C., F.G.M., J.C.C., C.R., I.R.S. conceived the study. A.C., J.C.C., C.R., S.L., F.G.M. performed data analysis. C.R., S.L., L.M., N.G. performed experimental work. All authors contributed to writing and editing the manuscript.

## Competing interests
The authors declare no competing interests.
