## [Peer Review File · Communications Biology]

Reviewers' comments:

Reviewer #1 (Remarks to the Author):

The functional methylome of symbiotic arbuscular mycorrhizal fungi reveals their unique place at the crossroads of the fungal tree of life, methylation of genes important to the symbiosis and among-nucleus methylome diversity

This manuscript describes the 6mA DNA methylation patterns which can be obtained from PacBio sequencing reads for *R. irregularis*. This fungal species is a well-known member of Early Diverging Fungi (EDF) and an obligate symbiont of land plants, believed to have played a critical role in the transition of photosynthetic organisms to land. *R. irregularis* has a well-characterised multi-nucleate spore that often contains thousands of nuclei. This makes this fungus very difficult to work with experimentally and, until long-read sequencing, made genome assembly very difficult. In this study the authors seek to avoid some of this genome heterogeneity by using several well-characterised (and commercialised) strains which are homokaryons. They also include two dikaryon strains which carry two unique nuclei in the same cytoplasm. In this work the authors seek to correlate 6mA methylation with gene expression and make inferences on how differential methylation could play a role in the symbiosis with plants. Unfortunately the methods used to obtain this data are flawed and severely compromise the Authors' ability to infer these relationships. This is because methylation is a dynamic mark which should be taken into account in the experimental design, starting with the conditions that the fungi were grown through to the storage of the DNA before sequencing.

For these reasons I believe the conclusions in the manuscript are vastly over-stated and significant revisions and or additional methylation data estimated from carefully controlled samples is required before consideration for publication.

Major Comments

The authors correlate the presence of 6mA to gene expression using epigenome and transcriptome data coming from different studies. It is unclear under which growth conditions the RNA-seq data were generated. Were these similar to the growth conditions from which the DNA was extracted? In general, DNA methylation data and RNA-seq data should be taken from the exact same sample in order to assess what impact methylation may have on gene-expression.

The 6mA data inferred from SMRT sequencing come from samples that were stored at 4°C before sequencing. How long were the fungal samples stored for was this treated equally for all strains? It appears some strains were grown commercially, while others were grown "in-house" by the authors. Storage can have an impact on the distribution of 6mA. For a meaningful correlation, RNA and 6mA data need to come from comparable growth conditions. To draw any conclusions about the actual effect of 6mA on gene expression, different conditions need to be tested where changing 6mA levels results in corresponding changes in gene expression.

5mC estimates from the mass spectrometry data are extremely high. While I recognize that 5mC cannot be estimated from PacBio data, the estimates provided here from mass spectrometry are extreme. The author report here 5mC methylation rate of 10-30% of the genome. However, the highest rate estimated genome-wide for other fungi appears to be in the range of 10-12% in Basidiomycetes, <https://doi.org/10.1038/s41559-019-0810-9>. Have the authors missed a real discovery here if the cytosine methylation rate is really that high? This is really extreme given that this is a very GC poor genome. Or is there something amiss with the mass-spectrometry? This could only be confirmed through bi-sulfite or potentially Nanopore sequencing. We note that the above experimental recommendations for sample growth, storage and subsequent sequencing for determining methylation should be strictly followed if 5mC is estimated.

The Genome assembly data is completely absent from Results which give the reader little confidence that genome coverage was actually sufficient to call methylation without digging through supplementary. I note that the dikaryon strain C3 has an interesting bi-modal distribution in the coverage estimates. This is perhaps an indication that there are quite unique regions of the

genome that are present in only one of the haploid nuclei. Were any attempts made to phase the genome assembly, so that this could be assessed? Would also be very interesting with a phased genome assembly to look at differential methylation between the two different haploid nuclei.

Below you will find some minor comments that may also help improve future drafts of this work.

Abstract

Lines 30-31: "Unlike Dikarya, 6mA methylation in AMF specifically occurs at symmetrical ApT motifs in genes ensuring their inheritance during DNA replication"
Wasn't this also shown by Mondo et al. 2017?

Background

Line 42 – 6mA is probably not the most important distinction between EDF and Dikarya?

Which DNA methyltransferases (5mC and 6mA) are present in EDF and Dikarya? Are there similarities which could be used formulate a hypothesis as to why *R. irregularis* might be more similar to Dikarya than other EDF?

Line 60-61- what about Nitrogen? Is phosphate the only nutrient that these fungi supply to plants?

Line 64 their- there

Line 71-75: How do these different clones influence rice or cassava growth? Are they all beneficial, just to different degrees? Are any detrimental?

Line 87- " Isolates C3 and A5 are dikaryons" Short sentence that seems out of place

Results

Lines 92-96: Need to provide more information rather than just refer to supplementary.
Assembled three isolates into XX chromosomes with an average coverage of XXX
How did the assemblies differ for the haploid homokaryon versus the dikaryon (did you phase the two haploid genomes?)

Lines 97-98: What is considered an SV? It seems from MM that very small 30bp deletions were included? Do you think small syntentic gene/presence absence is an SV? Or larger movements such as inversions/translocations?

Line 102: How do you define a clone? 95% identity? 98%?

Line 103-104: Given the importance that coverage has on being able to accurately call methylation, this seems like it should be in the main text

Line 109: 30 – 50% methylated cytosines? This is way more than in other fungi. Where are these methylated sites, genes or TEs? Would require additional data to assess this.

133: Genes harbouring 6mA – It is unclear from the methods what the threshold was for calling a gene as 6mA. One 6mA site or more?

133-134: "Out of a 290 BUSCO core fungal gene set, 270 of those genes occurring in *R. irregularis* were methylated" This seems like a high proportion of BUSCO genes looked at were methylated. If you compare this group to GO terms associated with Phosphate transport do you see a significant difference? i.e. are the genes related with symbiosis significantly higher methylated than a random set of core genes? This also relates to Figure 4. Can you more explicitly compare this with another key process that is not necessarily related to symbiosis and show a difference?

Line 175: "under regulation of 6mA" – there is no evidence that 6mA actually regulates gene expression here, only correlations.

In general, regarding regulation of genes by 6mA: All of this is speculative, there is no evidence presented that 6mA is involved in that. There needs to be either some evidence that 6mA levels change under different conditions or that deletion of the 6mA DNA methyltransferase affects gene expression.

Also, 6mA data and RNA-seq data do not come from the same experiment/conditions. This makes it very difficult to draw conclusions on effects of 6mA on gene expression (Fig 3C).

Lines 195-198: "Finally, the parts of the of the PKA signalling pathways, that are strongly upregulated in AMF in response to low P environments, and the MAPK and Tor signalling pathways, that are down-regulated in low P environments appear to be almost completely methylated, suggesting strong epigenetic control on the regulation of these pathways in the fungus in varying P environments."

Again you cannot conclude this with any certainty without measuring a change in 6mA between these different environments

Line 200-201: "while clearly having lost 6mA methylation from a very large number of other genes is highly intriguing"

Never stated here what percentage of genes are not methylated.. so difficult to see how the authors came to this conclusion

Fig 4: The font is very small and (almost) unreadable.

Methods

Line 230: DNA extraction: How does storage at 4°C affect 6mA? How long were the cells stored and was this different for the different isolates?

Line 337: RNA-seq analysis: Tophat and Cufflinks: It seems outdated to use these programs.

Line 330: "we only considered genes to be methylated which harbour ApT di-nucleotide motif" Is ApT the only context in which methylation can occur? And if not, why only analyse these sites?

Data accession is not available.

Reviewer #2 (Remarks to the Author):

This manuscript explores structural variations and the epigenomic landscape of *Rhizophagus irregularis*, a model organism for understanding the Glomeromycotina (arbuscular mycorrhizal fungi). To uncover DNA modifications, they use PacBio sequencing and identify symmetric 6mA at ApT dinucleotides, primarily at the starts of genes. They show key symbiosis genes that retain 6mA methylation, despite reduced total 6mA compared to most other early diverging fungi. While I believe this is an important discovery and will be of great value to the mycorrhizal community, there are several major concerns with the manuscript that need to be addressed before it is ready for publication. The most important of which is the narrative of the text, which puts forward the idea that AMF are somehow unique in their 6mA patterns and represent some transitional state between EDF and Dikarya. There are two problems with this narrative: 1) phylogenetics clearly places AMF in a monophyletic group within the Mucoromycota and not as a separate branch leading to the evolution of the Dikarya and 2) most of the data the authors present suggests that 6mA patterns in AMF are the same as in other early-diverging fungi. While 6mA abundance is on the lower side when considering % adenines methylated and # genes methylated, lower abundances than these were reported in Mondo et al., 2017 for other EDF lineages. Additionally, heterogeneity in methylation ratio (albeit not as high as here) as well as presence of both 6mA and 5mC within the same genome were both reported there as well – see major comment 1 for details. That said, the demonstration of 6mA presence in AMF and describing the types of genes that are methylated in these fungi represents a major advance in our understanding of AMF and potential fungus-plant communication. I believe this aspect of the manuscript should be

emphasized and receive more attention throughout the paper.

Major concerns:

1) Based on what is reported here, Glomeromycotina do not show any global 6mA patterns that are different from what was reported previously in other early-diverging fungi - specific examples below. The narrative in the text and interpretation of the data seems to argue differently. This needs to be rewritten so readers are not misled. For example, I find it particularly misleading to title a section '6mA and 5mC methylation levels in arbuscular mycorrhizal fungi mimic the Dikarya not the EDF'. In addition to what is mentioned below, Mondo et al. 2017 were unable to detect any 6mA in Dikarya using MS, even though PacBio reported modification presence. This is distinctly different from AMF, where the authors were able to confirm 6mA presence using both methods. Throughout the text, the authors should highlight the similarities with other EDF rather than try to make them seem different.

a) Low abundance (but clearly present) 6mA was reported by Mondo et al., 2017 in both *Linderina pennisporea* (Zoopagomycota) and *Rhizoclostridium globosum* (Chytridiomycota). For this paper on 6mA in AMF, the abstract highlights the reduced # of methylated genes in AMF compared to other EDF, yet *Linderina* has fewer methylated genes than AMF (~10% in *Linderina* [Mondo et al., 2017 supplementary figure 6] versus ~22% in AMF). Like what is reported here for AMF, *Linderina* also clearly has both 6mA and 5mC methylation.

b) *Rhizoclostridium* was only mentioned as a case where 6mA was low abundance and has not 'retained functional ApT motif methylation' (lines 154-155). Fig 1c from Mondo et al., 2017 indicates low but clear symmetric ApT methylation in this organism. This suggests that even if levels are low, it is possible that there are some functional and biologically meaningful 6mA modifications in this organism, even if their paper did not explore this further.

c) Lines 129-143: This is the same pattern observed in Mondo et al., 2017, where 6mA was most often observed at conserved genes in EDF (for example supplementary table 2).

d) Lines 28-29: Heterogeneity in methylation ratios are observed in other EDF, for example *Lobosporangium transversale* (Mondo et al., 2017; supplementary figure 1d), although not to the level observed in AMF. Again, while not discussed in their paper, it is important to mention this organism here to provide appropriate context, especially since *Lobosporangium* (Mortierellomycotina) is also a member of the Mucoromycota (and some phylogenies place Mortierellomycotina as sister to Glomeromycotina).

2) Something seems wrong with the phylogeny in figure 1. The topology is incongruous with most fungal phylogenies produced to date. In Figure 1, the Chytridiomycota and Blastocladiomycota appear as sister to the Dikarya, whereas in most phylogenies the Mucoromycota are sister to Dikarya (for example, see review by James et al., 2020:

<https://www.annualreviews.org/doi/full/10.1146/annurev-micro-022020-051835>).

3) I cannot find any methods describing how 5mC was detected, except for in Figure 1b where 5mC abundance is mentioned to come from LC-MS data. This needs to be added to methods.

4) Much of the supplementary material is devoted to structural variant analysis but this is barely mentioned in the main text (only in the 1st section of results where it is merged with description of genome coverage and annotation). I would recommend either removing SV analysis from the manuscript or significantly expanding on it in the main text and explaining how this contributes to our understanding of Glomeromycotina biology.

5) Lines 132-133 and 334: I am concerned that the threshold of a single modification is too low to consider a gene as methylated. I wonder if such low abundance is biologically meaningful, especially in AT rich organisms, like *R. irregularis*, where there is a higher chance of detecting methylated AT dinucleotides by chance. I think 6mA clusters should first be determined before calling genes that are methylated. Alternatively, perhaps genes can be called as 'methylated' if more than a certain minimum # of methylated ApTs (determined based on your data) are present at promoters.

6) The title of this manuscript is too long and unclear. It needs to convey a specific, succinct message. I also suggest removing language like 'reveals their unique place at the crossroads of the fungal tree of life' for reasons stated already.

7) Lines 137-143: This is a very interesting part of the paper and could have important implications for the mycorrhizal community. I recommend expanding on the genes that are differentially methylated between genetically identical isolates - what genes are they, and can any explain the differences these isolates have on observed crop growth? Same with differential methylation between nuclei in the dikaryon. I see details on these in the supplementary text -

they should be moved to the main text I think.

Minor comments:

1) Line 140-141: Language needs to be carefully chosen here - 6mA was shown to be associated with actively expressed genes in fungi previously, but not a pre-requisite for gene expression, since unmethylated genes can also be expressed at high levels, as you see both here and in Mondo et al., 2017.

2) Lines 20-21: "450 million years ago the EDF sub-phylum Arbuscular mycorrhizal fungi (AMF; Glomeromycotina)" Reword, this is confusing. Should be something like: "450 million years ago the Glomeromycotina (Arbuscular mycorrhizal fungi; AMF), a sub-phylum within the Mucoromycota"

3) Line 40-41: 1bya refers to the origin of fungi, and needs references.

4) Line 45: What does 'assuring active transcription' refer to in the context of 6mA? Again, like minor comment 1, language around 6mA's role in gene expression needs to be chosen carefully.

5) I noticed several places in the text where the authors were comparing AMF to 'EDF', yet AMF are clearly part of this group. They should compare AMF to 'other EDF'.

6) Figure S2 - strand is hard to see in the plot, consider either merging strands together since they show the same distribution, or using a different way to visualize them.

7) Lines 274-277: this section is unclear and I am having a hard time figuring out what the authors did, how genes were identified for tree building, and how many genes were used.

Reviewer #3 (Remarks to the Author):

Arbuscular mycorrhizal (AM) fungi are the ubiquitous symbiotic fungi of plants and promote global phosphate cycling in terrestrial ecosystems. Despite their biological importance, the genomic basis for gene expression regulation has hardly been clarified, because they are coenocytic and unculturable fungi.

The authors found the following points by using the Pacific Biosciences RSII platform with SMRT cell technology.

- AMF are systematically classified as EDF, but the abundance of 6mA is as low as 0.1 to 0.2%, which is different from other EDFs.
- On the contrary, 5mC, which is present only low in other EDFs, is detected frequently (~ 30%).
- The feature that 5mC is high is rather close to the feature of Dikarya.
- However, although the abundance of 6mA is low, they show a heterogenous pattern among isolates and may be involved in epigenetic heterogeneity and population-level adaptation.
- 6mA are localized in the core fungal genes.
- There is a symmetric pattern at the location of 6mA, which is similar to EDF and is likely to be maintained coupled to DNA replication.
- Enrichment from GO analysis to fungal hydrolase activity and transporter activity can be seen.

-This is positioned as an important study as an analysis of AMF that was not included in the previous 6mA analysis of EDF.

-It is an important achievement to clarify the characteristics of low 6mA abundance and high 5mC abundance, which is different from general EDF.

-It is interesting to see the difference in the pattern of 6mA among isolates from the viewpoint of causing non-uniform effects on plant growth.

I think these results are highly appreciated, as providing the basis for genome-wide gene expression of AMF.

To strengthen the message of this paper, it is desirable to perform the following analyses.

A 6mA difference among isolates has been detected. It seems that there are differences in the relevant methylase genes among isolates. It is advisable to identify the methylase genes from their genomes and show the differences among isolates.

AMF show the presence of high 5mC (Fig. 2b), but it would be even better if there were analytical

data comparing it with Dikarya to see where they are located in the genome.

Correlation analysis between expression level of genes of interest of isolates by qRT-PCR and 6mA pattern.

Finally, I think the following subheading does not have sufficient evidence.

"6mA methylation in AMF has a functional role in gene regulation similar to EDF"

Reviewer #4 (Remarks to the Author):

The article described six arbuscular mycorrhizal fungi (AMF) genome assemblies and the epigenome landscapes. In this study, authors found that unlike the other early-diverging fungi (EDF), AMF was more like Dikarya and other eukaryotes, which exhibited relatively low content of 6-methyldeoxyadenine (6mA) but high 5-methylcytosine (5mC) content. The author focused on analyzing the distribution of 6mA in the genome and its relationship with gene expression. They found that the genes containing 6mA are related to the character of arbuscular mycorrhizal fungi, such as phosphate metabolism and DNA methylation. These results were novel. However, after reading the whole story, I am confused by some results.

First of all, I think the main contribution of this work is providing six high-quality fungal genomes. I suggest that the authors compare the six de novo assembled genomes with other published fungal genomes, such as the orthologous genes or homologous genes, or the PAV or CNV of AMF genomes to the other EDF genomes. I believe that the biggest difference between AMF and other EDFs should be genetic differences. And, epigenetic differences may help increase the adaptability of AMF to different environments.

Second, the authors emphasize that the novelty finding of this study is the relatively low content of 6-methyldeoxyadenine (6mA) but high 5-methylcytosine (5mC) content in AMF genomes. For this result, I think it will better to use several representative EDF and Dikarya genome's methylation contents to highlight the methylation pattern of AMF. In addition, how to detect and calculate 5mC in this research? Since the 5-mC content in the AMF genome is high, why don't the authors pay more attention to 5mC?

Thirdly, how the association between DNA methylation and gene expression were concluded in this study? I notice in the last two or three paragraphs, the authors described that genes contained 6mA were under the regulation of methylation of 6mA. Generally, if we want to say DNA methylation regulates gene expression, we need to get the methylation mutant or use different experimental conditions to prove that relationship. So, I think it is not stringent to conclude that genes were regulated by 6-mA, only base on the fact that genes contain 6mA methylation.

In the last, it is an interesting result that genes contained 6mA were related to DNA methylation. Can authors provide more detailed information on this result, for example, how about the similarity of these genes to well-known methyltransferases in animal and plants?

Reviewer #1:

Major comments

The functional methylome of symbiotic arbuscular mycorrhizal fungi reveals their unique place at the crossroads of the fungal tree of life, methylation of genes important to the symbiosis and among-nucleus methylome diversity.

This manuscript describes the 6mA DNA methylation patterns which can be obtained from PacBio sequencing reads for *R. irregularis*. This fungal species is a well-known member of Early Diverging Fungi (EDF) and an obligate symbiont of land plants, believed to have played a critical role in the transition of photosynthetic organisms to land. *R. irregularis* has a well-characterised multi-nucleate spore that often contains thousands of nuclei. This makes this fungus very difficult to work with experimentally and, until long-read sequencing, made genome assembly very difficult. In this study the authors seek to avoid some of this genome heterogeneity by using several well-characterised (and commercialised) strains which are homokaryons. They also include two dikaryon strains which carry two unique nuclei in the same cytoplasm. In this work the authors seek to correlate 6mA methylation with gene expression and make inferences on how differential methylation could play a role in the symbiosis with plants. Unfortunately the methods used to obtain this data are flawed and severely compromise the Authors' ability to infer these relationships. This is because methylation is a dynamic mark which should be taken into account in the experimental design, starting with the conditions that the fungi were grown through to the storage of the DNA before sequencing.

For these reasons I believe the conclusions in the manuscript are vastly over-stated and significant revisions and or additional methylation data estimated from carefully controlled samples is required before consideration for publication.

A1. We thank the reviewer for appreciating the novelty of our study and we address the specific concerns below.

The authors correlate the presence of 6mA to gene expression using epigenome and transcriptome data coming from different studies. It is unclear under which growth conditions the RNA-seq data were generated. Were these similar to the growth conditions from which the DNA was extracted? In general, DNA methylation data and RNA-seq data should be taken from the exact same sample in order to assess what impact methylation may have on gene-expression.

A2. The growth conditions for both RNA-seq and DNaseq (long-read sequencing) were exactly the same. While it is true that 6mA marks in the genome could be dynamic, at least all the samples have been treated in the same way and grown in exactly the same conditions. As the reviewer will see, our results indicate that 6mA in these fungi may well not be very dynamic. Firstly, there is a very high conservation of which genes are methylated across the different isolates (actually this is consistent with conserved 6mA marks in EDF in general). This wouldn't be expected if the marks were very dynamic and if small differences in growth conditions would lead to lots of changes in 6mA methylation marks.

Secondly, while RNA was extracted from a different set of material to the DNA (but grown in exactly the same conditions), if this would make a difference, we wouldn't expect to see an association between methylated genes and gene transcription levels. If cultivating the fungi at different times really affected the presence of methylation marks, we would not expect such an association. Actually, we consider the fact that we see an association between gene expression levels and 6mA methylation marks on material grown at different times actually strengthens the argument for this association. We suspect that had we done this at exactly the same time with the same material we would have just observed stronger associations.

The 6mA data inferred from SMRT sequencing come from samples that were stored at 4°C before sequencing. How long were the fungal samples stored for was this treated equally for all strains? It appears some strains were grown commercially, while others were grown “in-house” by the authors. Storage can have an impact on the distribution of 6mA.

A3. We only report 6mA data on 3 of the isolates; namely, C2, C3 and C5, and not all 6. We did not have enough coverage of B12, A5 and A1 to record 6mA marks. Only A1 was grown separately from the others in house rather than at Symbiom. C2, C3 and C5 were produced in identical conditions by Symbiom (Lanškroun, Czech Republic) and, thus, represent uniform culture conditions for all three isolates. Symbiom did not produce the fungi in their commercial production system but in identical conditions to ours. We have worked many years with this company to produce material for us in exactly the same conditions as they have greater manpower to do that than in our group but this is not a commercial formulation or commercial venture or a different preparation. All spores were harvested and stored at 4°C for the same length of time. Again, if this had had an effect on presence of 6mA marks then we wouldn't expect to see the observations we have made such as conservation of genes containing 6mA marks or the association with gene expression.

For a meaningful correlation, RNA and 6mA data need to come from comparable growth conditions. To draw any conclusions about the actual effect of 6mA on gene expression, different conditions need to be tested where changing 6mA levels results in corresponding changes in gene expression.

A4. We do not claim to have experimentally observed a direct effect between the presence of an 6mA mark in the genome and gene expression: We claim that it is highly likely that 6mA marks in AMF regulate gene expression. This is based on several pieces of complimentary evidence which makes the argument indeed compelling. These are: 1) There is an association between the presence of 6mA marks as symmetrical ApT motifs in a gene and the expression of the gene (as observed for other EDF by Mondo et al.) that suggests that 6mA marks are implicated in upregulation of those genes. 2) The genes in which the marks occur are conserved across isolates which would not be expected if the marks had no function. 3) The vast majority of the marks are at symmetrical ApT motifs which are a feature of experimentally demonstrated epigenetic marks in the genomes of other eukaryotes. Again, it would be highly unlikely that selection would ensure the conservation of ApT motifs if they did not have a functional role. 4) The 6mA ApT motifs are almost exclusively occurring in gene bodies, i.e., exactly at the place where they would be expected if they had a functional role in regulating gene expression. Taken together the association between the marks and

expression and the motifs and positions of the marks it is highly likely that 6mA regulates gene expression.

For the comment about comparable growth conditions please see answer A3 and the 2nd part of our answer A2.

5mC estimates from the mass spectrometry data are extremely high. While I recognize that 5mC cannot be estimated from PacBio data, the estimates provided here from mass spectrometry are extreme. The author report here 5mC methylation rate of 10-30% of the genome. However, the highest rate estimated genome-wide for other fungi appears to be in the range of 10-12% in Basidiomycetes, <https://doi.org/10.1038/s41559-019-0810-9>. Have the authors missed a real discovery here if the cytosine methylation rate is really that high? This is really extreme given that this is a very GC poor genome. Or is there something amiss with the mass-spectrometry? This could only be confirmed through bi-sulfite or potentially Nanopore sequencing. We note that the above experimental recommendations for sample growth, storage and subsequent sequencing for determining methylation should be strictly followed if 5mC is estimated.

*A5. We agree with reviewers that the majority of fungal species studied to date have 5mC methylation proportions to non-methylated C in the range the reviewer mentioned. However, we observed a high 5mC methylation % in *R. irregularis* based a high-resolution mass spectrometry technique which is the gold standard for verifying methylation content. The observation was consistent among the technical replicates of each isolate. Moreover, the detected range among six isolates were very similar. LC-MS is probably the most accurate way of measuring the proportion of methylated to non-methylated nucleotides but is limited in that it gives no information about their positions. For detecting amounts of 5mC (but not their positions), LC-MS is considered far more accurate than bi-sulphite sequencing or Nanopore sequencing. Recently, some companies have tried to develop enzymatic conversion rather than bi-sulphite conversion (see New England Biolabs for example) because these sequencing methods don't accurately measure all 5mC because of the conversion. Even with the newer techniques, LC-MS is still considered the most accurate method. We are, thus, confident of our results. However, we apologise because during the various edits of our manuscript, the specific information in the methods about the standards used to verify 5mC proportions were removed. We have now added these to make it clear. The method we have used is really the best one out there, appropriate standards were used as well as sufficient replication.*

It's true that the percentage of 5mC is high. That's one of the points of our study that normally when 5mC levels are very high in an organism, 6mA is essentially absent or appears non-functional, which does not seem to be the case in AMF. The reviewer says that if that's the case, we have missed something. Not exactly. We are aware the % is high and very interesting when we also see likely function 6mA methylation but we can't really say more than that from our data. We had originally conducted Pac Bio with the intention of using this method to also detect 5mC marks. However, since we originally did the sequencing there were several papers indicating that Pac Bio data are not reliable for that. For this reason, we don't report them. However, the putative 5mC marks in the Pac Bio sequence data are also very high (if you can trust them).

The reviewer comments that the level of 5mC is surprising given that AMF genomes are relatively low in GC. We don't really follow this logic. It's important to realise that the LC-MS measurement of 5mC is just a percentage of methylated Cs to non-methylated C's and not total methylated Cs based on the amount of C in the genome.

The Genome assembly data is completely absent from Results which give the reader little confidence that genome coverage was actually sufficient to call methylation without digging through supplementary.

*A6. The genome data and details are available in the supplementary information. They are not directly reported in the main manuscript because it was not the focus of this study to report on the genome because the genome sequences of several *R. irregularis* isolates have already been sequenced. The important information for the reader is that the coverage of C2, C3 and C5 was high enough to call 6mA marks. We have added a sentence in the main manuscript stating the coverage for these isolates and obviously if the reader has some doubts and wants to check it in detail then can go into the supplementary information to verify this. However, it's true that it is the first report of long-range sequencing of a dikaryon of the *Glomeromycotina* (see next comment).*

I note that the dikaryon strain C3 has an interesting bi-modal distribution in the coverage estimates. This is perhaps an indication that there are quite unique regions of the genome that are present in only one of the haploid nuclei. Were any attempts made to phase the genome assembly, so that this could be assessed? Would also be very interesting with a phased genome assembly to look at differential methylation between the two different haploid nuclei.

A7. The reviewer is correct in his/her interpretation that we might see a bimodal distribution of coverage estimates for isolate C3 indicating the presence of unique regions in the genome of one nucleus genotype within this isolate. The reviewer is correct that there are unique regions of the genome that are present in each of the two nucleus genotypes. This is another focus in our group. However, here we focused on the methylation part because there are already plenty of published studies that describe AMF genomes that compare them thoroughly among AMF and across fungal phylogeny. Although we initially considered to attempt the phasing of the dikaryotic genomes, we could not because we lacked the Hi-C sequencing data necessary to resolve the haplotype switches. Instead, we found a compromise by using the diploid-aware FALCON-Unzip assembly software to identify potential heterogeneous regions in these genomes. This software produces a partially phased primary assembly and a secondary haplotig file (genomic regions with sequence differences between the two haplotypes). This is reported in the manuscript and this allowed us to then see if there is differential methylation on the two nucleus genotypes.

A more detailed explanation of this method can be found in the FALCON-Unzip article/github, where it is shown and explained why it is not possible to resolve the haplotype switches only using long-read sequencing data (chin et al 2016; PMID:27749838).

Minor comments

Abstract

Lines 30-31:“Unlike Dikarya, 6mA methylation in AMF specifically occurs at symmetrical ApT motifs in genes ensuring their inheritance during DNA replication”
Wasn't this also shown by Mondo et al. 2017?

A8. The previous study of Mondo et al. 2017 did see this in EDF but didn't investigate AMF. We found that AMF also showed symmetrical ApT methylation as other EDF reported in Mondo et al. 2017, even though AMF have lost most of their 6mA methylation compared to other EDF and have evolved to have a high proportion of methylated C. Our study that shows that AMF are unusual by having what look like functionally active 6mA marks (like in EDF) but also having high levels of methylated C.

We have greatly modified the abstract since the first submission as this journal expects a short abstract of not more than 150 words. So, the sentence the reviewer refers to is different now and the abstract is unfortunately now less specific. This is just to conform to the journal style.

Background

Line 42 – 6mA is probably not the most important distinction between EDF and Dikarya?

A9. Indeed, we didn't intend to imply that this is the only difference between these groups. We agree with the reviewer that the two groups of fungi are fundamentally different in many aspects of their biology including their morphological and genomic features, not simply by the 6mA. However, as pointed by Mondo et al. 2017, it is clear that 6mA of genome is highly distinct between two groups and we intended to deliver this message. To improve clarity, now we modified the sentence in the revised version of the manuscript, but in the introduction as the abstract has to be shortened (see A8 above).

Which DNA methyltransferases (5mC and 6mA) are present in EDF and Dikarya? Are there similarities which could be used formulate a hypothesis as to why *R. irregularis* might be more similar to Dikarya than other EDF?

A10. This is a very interesting comment from the reviewer and also mentioned by other reviewers. Inspired by reviewers' comments, we have undertaken a bioinformatic analysis of methyltransferase (MTase) conservation in AMF and further compared this across the fungal phylogeny and this revealed very interesting results.

There is a substantial new section of text in the manuscript about this analysis, including a new main figure (Figure 4) and another in the supplementary information. We think that this addresses this interesting point raised by this reviewer and some of the other reviewers. We think that this greatly improve the manuscript and we are thankful to the reviewer(s) for raising this issue.

Line 60-61- what about Nitrogen? Is phosphate the only nutrient that these fungi supply to plants?

A11. We agree with the reviewer that these fungi can also supply various other nutrients to plants, although phosphate supply is by far the most studied and the clearest. In soils where nitrogen is more limiting (usually because of slow organic matter decomposition processes) plants usually form symbioses with other fungi (ectomycorrhizal fungi) that are better adapted for acquisition of nitrogen because of their saprotrophic capabilities. We focus here on the main reported function of this symbiosis, and that's why we have emphasized the description regarding phosphate. For a better overview, the book, The Mycorrhizal Symbiosis (Smith & Read, 2008) gives a good description of the contribution of AMF to acquisition of nutrients other than phosphorus, but if you read that book you will see that the vast focus of physiological studies has been on phosphorus. We improved the statement and noted that even though we mainly focused on phosphate, the fungi can support plant acquisition of various nutrients, including nitrogen

Line 64 their- there

A12. Changed

Line 71-75: How do these different clones influence rice or cassava growth? Are they all beneficial, just to different degrees? Are any detrimental?

A13. The references cited at this point show that rice and cassava growth can vary five-fold and three-fold, respectively, just by inoculating the plants with genetically different isolates or different progeny from one isolate. None have been shown to be detrimental in the conditions that we tested them, but they have not all been shown to be beneficial either. However, this depends on the environment in which they are tested (e.g., how much P there is in the soil), the plant genotype etc. It's too much to go into here in this paper as it is not the main focus, but we have cited the articles so that a reader has access to them.

Line 87- "Isolates C3 and A5 are dikaryons" Short sentence that seems out of place

A14. This sentence is just to explain that two isolates among six sequenced isolates are dikaryons and is needed to understand the next sentence. If Pac Bio sequencing is conducted on a dikaryon, and if each nucleus is haploid, then variation among reads at a given locus represents differences between the two nucleus genotypes.

Results

Lines 92-96: Need to provide more information rather than just refer to supplementary. Assembled three isolates into XX chromosomes with an average coverage of XXX How did the assemblies differ for the haploid homokaryon versus the dikaryon (did you phase the two haploid genomes?)

A15. The focus of the manuscript is about m6A methylation. Hence this paragraph allows the readers to access the detailed technical information regarding genome assembly and its statistics in the supplementary information. The detailed information about the genome assemblies is important technical information if a reader wants to be assured that what we did was technically correct and that the data is of a good enough quality to be reliable.

However, going into lots of details about the genome assemblies in the main text greatly detracts from the main focus of the study about 6mA methylation.

We have answered the question about phasing in point A6 above.

Lines 97-98: What is considered an SV? It seems from MM that very small 30bp deletions were included? Do you think small syntentic gene/presence absence is an SV? Or larger movements such as inversions/translocations?

A16. Setting a length threshold for SV is rather arbitrary. It has traditionally been set at a minimum of 50bp, especially with Illumina data. Long-read sequencing allows the accurate targeting smaller variants and that's why we opted for a 30 bp threshold (as many other studies have done). These smaller variants are, by definition, also structural variants.

An article in Nature Reviews Genetics (2018) states:

“The Genome of the Netherlands Consortium performed WGS of 250 healthy families and reported 0.16 de novo SVs per generation (>20 bp in length, in contrast to most other studies, which use >50 bp as a threshold for an SV).”

<https://www.nature.com/articles/s41576-018-0007-0>

Here, is an example of another study, recently (2020) published in Cell, where they used the exact same methods to call SV (NGMLR+SNIFFLES) and they clearly defined SVs > 30bp.

“Reads were aligned to the recently released SL4.0 reference genome (Heinz 1706, SLL) with NGMLR, and SVs were called with Sniffles (Figures SIC and SID; Hosmani et al., 2019, Sedlazeck et al., 2018a). We then filtered, sequence resolved, and merged all 100 sets of SV calls, revealing 238,490 total SVs (defined in this study as >30 bp) that comprise the most comprehensive sequence-resolved panSV genome in plants (see STAR Methods). Importantly, we confirmed that the majority of these variants would not have been revealed using solely short-read sequencing data (Figure S1E).”

<https://www.sciencedirect.com/science/article/pii/S0092867420306164?via%3Dihub>

So, while the chosen threshold seems arbitrary, the threshold we used is in-line with contemporary studies using long-range sequencing to detect SVs. We have now indicated this in the methods.

Line 102: How do you define a clone? 95% identity? 98%?

A17. We have changed this sentence slightly to “are likely clones”. Actually, it is impossible to say with 100% certainty that any two individuals are absolutely 100% identical as some small differences can always occur for technical reasons, even if they are truly identical. So, because of this we say “likely clones”.

Based on short-read sequencing data, previous studies failed to observe differences in their haplotypes indicating that C2 and C5 were genetically identical (reference Wyss et al 2016 PMID: 26953600). Here, we investigated the structural differences between the genomes of C2 and C5 (that couldn't be targeted with short-read sequencing). We observed that these

two isolates shared almost all their SVs. Manually curating the putative variants revealed that they were unlikely to be true SVs. This is the reason we refer to them as likely clonal isolates. (This is also discussed in detail in Savary et al. 2018 ISME Journal).

The SV, or lack of, between C2 and C5 is described in the text of the supplementary information with the following text:

“No clear differences could be observed in SV between isolates C2 and C5. After manually curating putative isolate-specific variants against alignment files in IGV, it was clear that these variants were differentially called as a result of low read support values and/or insufficient coverage. These were, therefore, disregarded (SI Appendix Figure 4). Thus, these results support the hypothesis that these R. irregularis isolates are likely clones.”

Line 103-104: Given the importance that coverage has on being able to accurately call methylation, this seems like it should be in the main text

A18. We have now added the information “Sufficient coverage (25x per strand) allowed reliable detection of 6mA in C2, C5 and C3 (SI Appendix Fig. S2).”

Line 109: 30 – 50% methylated cytosines? This is way more than in other fungi. Where are these methylated sites, genes or TEs? Would require additional data to assess this.

A19. This is discussed above at point A5 in this document. The standard LC-MS protocol was followed to identify the methylated cytosines. The methodology is now fully described in manuscript. Since the technique used to measure the level of methylated cytosine was LC-MS, there is no sequence level information. For more clarity we added “High prevalence of 5mC methylation than other fungal species, suggests an important role of this epigenetic mark in gene regulation in AMF and requires further molecular characterisation.”

133: Genes harbouring 6mA – It is unclear from the methods what the threshold was for calling a gene as 6mA. One 6mA site or more?

A20.

1) In our analyses, we first counted the number of genes affected by methylation that even harboured a single ApT di-nucleotide motif. Out of a total of 7062 genes that carried one or more ApT motifs, 4398 genes (62%) were common to all three isolates (C2, C5 and C3). Of those 4398 genes, the mean number of ApT di-nucleotide motifs per gene was approx. 10 but with a distribution skewed to the right (see the new figure - supplementary figure S7). While we included them in our analysis, the number of genes considered methylated but only containing a single ApT motif were negligible (see supplementary figure S7). Furthermore, out of 244 (5.54% of 4398 genes), 290 (6.59% of 4398 genes) and 225 (5.11% of 4398 genes) genes that only harboured a single ApT di-nucleotide motif, in C2, C5 and C3 respectively, only 75 genes (1.70% of 4398) in C2, 89 genes (2.02% of 4398 genes) in C5, 68 genes (1.54% of 4398) in C3 were not consistently methylated at a given site in all three isolates.

2) Similarly, only very low percentages of single ApT di-nucleotide motif sites could not be validated in the commonly methylated genes between two isolates: (a) 1236 genes were commonly methylated between C2 and C5. Of these, 156 genes in C2 had a single ApT di-nucleotide motif but only of those 15 genes (1.21% of 1236 genes) were not methylated at the

same position in C5. In C5, out of 194 genes with single ApT di-nucleotide motif, only 14 genes (1.13% of 1236 genes) were not methylated at the same position in C2. (b) 314 genes were commonly methylated between isolates C2 and C3. Of these, 77 genes in C2 had a single ApT di-nucleotide motif and 34 were not methylated at the same position in C3. In C3, among 51 genes with single ApT di-nucleotide motif, 16 genes were not methylated at the same position in C2. (c) 149 genes that were commonly methylated between isolates C5 and C3. 44 of these genes in C5 had a single ApT di-nucleotide motif but only 15 genes were not methylated at the same position in C3. In C3, among 28 genes with a single ApT di-nucleotide motif, only 5 genes were not methylated at the same position in C5.

Overall, we found that the majority of the genes methylated at only a single ApT di-nucleotide motif occurred in two or more replicates with little differential methylation at the sites which were not methylated with the same and single ApT di-nucleotide motif.

133-134: “Out of a 290 BUSCO core fungal gene set, 270 of those genes occurring in *R. irregularis* were methylated” This seems like a high proportion of BUSCO genes looked at were methylated. If you compare this group to GO terms associated with Phosphate transport do you see a significant difference? i.e. are the genes related with symbiosis significantly higher methylated than a random set of core genes? This also relates to Figure 4. Can you more explicitly compare this with another key process that is not necessarily related to symbiosis and show a difference?

A21.

This comment is a very interesting and important one that we have considered carefully. The reviewer actually raises several different issues in this one comment and suggests an analysis to address this that we don't think is straightforward or possible with the current dataset.

The 3 issues in the question are:

- 1. The reviewer says that methylation in the core set of genes is very high (this first comment is not about symbiosis genes or P transport). What the reviewer points out is, indeed, true. The core set is surprisingly high. It's true that in other EDF, the core fungal gene set were also heavily methylated. However, those fungi had much higher levels overall of 6mA methylation than *R. irregularis*. In *R. irregularis* 22-23% of all genes were methylated and yet 93% of the core set of fungal genes were methylated. So, actually there is an enrichment of methylation in the core set of genes. The “core set” represents a set of genes that define the kingdom Fungi. The fact that these genes have enriched methylation suggests that 6mA methylation has been retained in a fundamental set of very well conserved genes necessary for fungal life. This is now stated more clearly in the manuscript.*
- 2. The second issue is about comparing the core set to GO terms associated with P transport. This analysis isn't really possible. Our gene ontology analysis really answers the question the reviewer is asking as genes representing the GO terms associated with transport are more enriched than genes falling into many other GO terms. However, the core set of fungal genes would not appear as a GO term themselves. These separate analyses that have been done, taken together say that*

6mA methylation has been retained in a set of well conserved genes fundamental to fungal life, as well as certain groups of genes representing different processes such as transport activity, methyltransferase activity etc. The transporter activity GO term is interesting in terms of symbiosis because AMF transport nutrients. But they transport many things. Fig 5c shows the transport activity in greater detail and only a small part is about P, the rest is about other transporters. So, the only thing we can say from this is that this large group is significantly enriched.

- 3. We have specifically picked out the very small number of genes that are known to be involved in the mycorrhizal symbiosis and in P homeostasis in the fungus. There are only a small handful of genes in the fungus that have been identified as being involved in the symbiosis. However, while this is actually a really small group of genes, these genes are methylated with 6mA. It's interesting that they are all methylated but the group is far too small to do any comparative analysis with a core set. We can't just take all genes involved in P transport (or a larger group for any sort of transport) because those genes are not identified as being involved in the symbiosis.*

Line 175: “under regulation of 6mA” – there is no evidence that 6mA actually regulates gene expression here, only correlations.

A22. 6mA mediated regulation of gene expression in eukaryotes when occurring at symmetrical ApT motifs has been reported (Li et al 2019, PMID: 31208067) and the association between up-regulation and 6mA marks at ApT motifs was shown in EDF (Mondo et al., 2017). We have answered this question above (see A4). It is true that there is no direct experimental evidence to demonstrate that in AMF 6mA regulates gene expression. However, the comparison of 6mA methylation of genes and corresponding transcriptome abundance of same isolate cultured in same condition showed clear positive correlation. The observed correlation is clearly showing that if a gene expression is more abundant then the gene is more likely to be 6mA methylated. The same pattern was observed in all isolates and suggests there is a relationship between 6mA methylation and corresponding gene expression. Coupled with the other evidence about the non-random pattern of methylation marks, and the fact that they occur in ApT motifs (a 6mA methylation signature known to regulate gene expression in some other eukaryotes) it is highly likely that these marks are involved in regulation of gene expression. We have improved the clarity of sentences.

In general, regarding regulation of genes by 6mA: All of this is speculative, there is no evidence presented that 6mA is involved in that. There needs to be either some evidence that 6mA levels change under different conditions or that deletion of the 6mA DNA methyltransferase affects gene expression.

Also, 6mA data and RNA-seq data do not come from the same experiment/conditions. This makes it very difficult to draw conclusions on effects of 6mA on gene expression (Fig 3C).

A23.

We agree that, like Mondo et al., our study only shows an association between 6mA marks and gene expression and the likelihood that the marks are involved in gene expression because of where they occur. We are not currently able to knockout genes in AMF and the multi-nucleate state of these fungi has rendered transformation unsuccessful with these fungi

which prevent the sort of functional studies mentioned by the reviewer. Actually, the role of 6mA marks in EDF, based on Mondo et al.'s findings which are also based on association have also not been challenged because of this. However, we consider that the cumulative evidence we present provide a strong argument that these marks are highly likely to be functionally significant in regulating gene expression. If these marks did not play some function in gene regulation, then why would they almost always be located in the same region of a gene body? Why would they be located in motifs known to play a role in gene regulation in other eukaryotes? Why would they be conserved across genetically different isolates? Why should they be correlated with gene expression levels? We would expect none of these features to have been conserved during evolution unless they had some functional significance that could ultimately affect fungal fitness through effects on the phenotype.

The reviewer says that it is inappropriate to do the genome sequencing on one set of material and the RNAseq on material grown at a different time. We agree with the reviewer that it would ideally be done at the same time on the same material and we would have done that if material had allowed this. However, it would also be very hard to explain why gene expression levels in several replicates of three genetically different isolates would all be associated with the presence of 6mA signatures in genes just by chance. Each correlation has a probability indicating the likelihood that the association occurred by chance. But what is the chance that each replicate of each isolate would all show the same correlation just by chance? We agree that the evidence is observational rather than experimental, but the likelihood of there not being an association seems very weak.

We appreciate the reviewer's comments and we are well aware of these issues and because of that we have been very careful not to say there is a causal relationship but that cumulative evidence points to an association between methylation and gene expression.

Lines 195-198: "Finally, the parts of the of the PKA signalling pathways, that are strongly upregulated in AMF in response to low P environments, and the MAPK and Tor signalling pathways, that are down-regulated in low P environments appear to be almost completely methylated, suggesting strong epigenetic control on the regulation of these pathways in the fungus in varying P environments."

Again you cannot conclude this with any certainty without measuring a change in 6mA between these different environments

A24. The reviewer is indeed correct, and we have modified the text to "likely under epigenetic control". We meant to suggest that these pathways are probably under 6mA epigenetic control instead of claiming their dynamics in different in various environments.

Line 200-201: "while clearly having lost 6mA methylation from a very large number of other genes is highly intriguing"

Never stated here what percentage of genes are not methylated.. so difficult to see how the authors came to this conclusion

A25. We have now mentioned the percentage of methylated genes "Specifically, 22.8%, 22.8% and 21.9% of genes in C2, C5 and C3 respectively harboured 6mA." See line L132.

Fig 4: The font is very small and (almost) unreadable.

A26. Please note, because of the addition of a new figure 4, the figure we are referring to here is now Fig. 5

We have attached a high-resolution PDF file to solve this problem, so that hopefully this can be viewed by the reviewers and can be used online for the journal if the manuscript is accepted. However, because of the software used to make this figure, we lose resolution when we insert it into the manuscript into the Word version required by the journal.

We have also attached a higher resolution version of Fig S2, as well.

Methods

Line 230: DNA extraction: How does storage at 4°C affect 6mA? How long were the cells stored and was this different for the different isolates?

A27. C2, C5 and C3 were all maintained and stored in the same manner for the same time at 4°C. It is unknown how, if at all, refrigeration would affect AMF spores or their 6mA methylation. The intent of this study was not to answer how temperature affected methylation, and therefore 4°C was maintained constant for all lines. So, while we appreciate the reviewer's comments, and agree temperature may be a factor in methylation of this species, we feel we have taken the appropriate steps in standardizing conditions across the individuals compared.

Line 337: RNA-seq analysis: Tophat and Cufflinks: It seems outdated to use these programs.

A28. Both programs used in RNA-seq analysis have nearly 10000 citations each and they are still widely used. There are newer methods but it is unclear what the advantages would be. For example, we could have used HISAT2 but we still think Tophat and Cufflinks serve the purpose for this analysis.

Line 330: “we only considered genes to be methylated which harbour ApT di-nucleotide motif” Is ApT the only context in which methylation can occur? And if not, why only analyse these sites?

A29. We focused on ApT di-nucleotide motif as this was by far the most abundant di-nucleotide motif in all three isolates as described, 6mA marks in R. irregularis occurred as symmetrical methylation at ApT di-nucleotide motifs (73% in C3; 68% in C2 and C5). Also, we decided to more conservatively focus on the symmetric motif because if the both strands were called as methylated (X25 coverage per each strand), it is less likely to be due to erroneous detection. Moreover, the symmetrical methylation allows parental strands to carry marks through DNA replication as well as to propagate them to the newly synthesized child strand, thus, it is a biologically interesting motif that has been shown to have a function in some other eukaryotes.

There is very little known about function of 6mA at none ApT motifs.

Data accession is not available.

A30. The data has been submitted to a repository with the accession number given in the manuscript. It will be released on publication. This is fairly standard practice. However, if the reviewer would like to access the data to check some details, we are happy of course to allow access.

Reviewer #2 (Remarks to the Author):

This manuscript explores structural variations and the epigenomic landscape of *Rhizophagus irregularis*, a model organism for understanding the Glomeromycotina (arbuscular mycorrhizal fungi). To uncover DNA modifications, they use PacBio sequencing and identify symmetric 6mA at ApT dinucleotides, primarily at the starts of genes. They show key symbiosis genes that retain 6mA methylation, despite reduced total 6mA compared to most other early diverging fungi. While I believe this is an important discovery and will be of great value to the mycorrhizal community, there are several major concerns with the manuscript that need to be addressed before it is ready for publication. The most important of which is the narrative of the text, which puts forward the idea that AMF are somehow unique in their 6mA patterns and represent some transitional state between EDF and Dikarya. There are two problems with this narrative: 1) phylogenetics clearly places AMF in a monophyletic group within the

Mucoromycota and not as a separate branch leading to the evolution of the Dikarya and 2) most of the data the authors present suggests that 6mA patterns in AMF are the same as in other early-diverging fungi. While 6mA abundance is on the lower side when considering % adenines methylated and # genes methylated, lower abundances than these were reported in Mondo et al., 2017 for other EDF lineages. Additionally, heterogeneity in methylation ratio (albeit not as high as here) as well as presence of both 6mA and 5mC within the same genome were both reported there as well – see major comment 1 for details. That said, the demonstration of 6mA presence in AMF and describing the types of genes that are methylated in these fungi represents a major advance in our understanding of AMF and potential fungus-plant communication. I believe this aspect of the manuscript should be emphasized and receive more attention throughout the paper.

A31. We thank the reviewer for appreciating our work. In the revised version of the manuscript, we have implemented changes that clarify the doubts and answered them below.

Major concerns:

1) Based on what is reported here, Glomeromycotina do not show any global 6mA patterns that are different from what was reported previously in other early-diverging fungi - specific examples below. The narrative in the text and interpretation of the data seems to argue differently. This needs to be rewritten so readers are not misled. For example, I find it particularly misleading to title a section '6mA and 5mC methylation levels in arbuscular mycorrhizal fungi mimic the Dikarya not the EDF'. In addition to what is mentioned below, Mondo et al. 2017 were unable to detect any 6mA in Dikarya using MS, even though PacBio reported modification presence. This is distinctly different from AMF, where the authors

were able to confirm 6mA presence using both methods. Throughout the text, the authors should highlight the similarities with other EDF rather than try to make them seem different.

A32. The reviewer is correct that the subsection heading was not accurate. We have tried to make the text clearer to clarify the confusion in the manuscript as a whole. But since this is a major concern of the reviewer, here is our explanation why we say the Glomeromycotina methylation patterns are not exactly like other EDF but are also not exactly like the Dikarya:

Our claim that AMF show unique methylation patterns compared to EDF or Dikarya is not based on the levels of 6mA alone, or the level of 5mC alone, but a combination of the level of 5mC, the level of 6mA, the position of the 6mA methylation marks, and its heterogeneity.

*To summarize: Taken together, our data show that *R. irregularis* has a low % of 6mA marks in the genome but which are mostly occurring in gene bodies at symmetrical ApT motifs, it is heterogeneous, it is associated with active gene transcription but at the same time the level of methylated C is very high (between 30% and 50%). There are no EDF that jointly exhibit these characteristics, nor do any Dikarya appear to exhibit this combination of characteristics. This is what we mean when we say that AMF show a unique pattern of methylation not observed in other fungi. You could put it another way which would be that they share some methylation patterns of both EDF and Dikarya but are not entirely like either – hence the change to the title.*

*Most EDF studied by Mondo et al have high levels of 6mA. Indeed, the reviewer is absolutely correct that a small number of the EDF studied by Mondo et al. also exhibited low levels of 6mA, namely, *R. globosum*, *C. anguillulae* and *L. pennispora* (moderate –but still quite a lot higher than *Glomeromycotina*).*

*Of these, *C. anguillulae* is unlike AMF as it has almost no 6mA at ApT motifs. It has some 5mC but at levels that are very low compared to AMF. According to Mondo et al. Fig S8a it contains 0.5% of total cytosine).*

**R. globosum* has 6mA at ApT motifs but at a much lower level than in *Glomeromycotina* and only 0.2% 5mC.*

**L. pennispora* exhibits a considerably higher 6mA level than in AMF and an almost undetectable level of 5mC.*

*So, in summary, yes, the reviewer is correct that if you just look at one characteristic like the level of 6mA in *Glomeromycotina* and other EDF, there are some that show the similar levels. But they don't share some of the other important and notable features.*

I hope we have now clarified our claim in the manuscript.

a) Low abundance (but clearly present) 6mA was reported by Mondo et al., 2017 in both *Linderina pennispora* (Zoopagomycota) and *Rhizoclosmatium globosum* (Chytridiomycota). For this paper on 6mA in AMF, the abstract highlights the reduced # of methylated genes in AMF compared to other EDF, yet *Linderina* has fewer methylated genes than AMF (~10% in *Linderina* [Mondo et al., 2017 supplementary figure 6] versus ~22% in AMF). Like what is reported here for AMF, *Linderina* also clearly has both 6mA and 5mC methylation.

A33. Yes, we agree with reviewer that Mondo et al. reported low abundance for both Linderina pennisporea and Rhizoclostridium globosum. However, in case of Linderina pennisporea it does not have high 5mC methylation (Supplementary Figure 8 of Mondo et al.) making it a clear difference between AMF and this species. To accommodate the “fewer methylated genes” suggestion we have now updated our sentence in the abstract “While this reduction in 6mA methylation means that far fewer genes are regulated by 6mA in the AMF genome than in most EDF ...”

b) Rhizoclostridium was only mentioned as a case where 6mA was low abundance and has not 'retained functional ApT motif methylation' (lines 154-155). Fig 1c from Mondo et al., 2017 indicates low but clear symmetric ApT methylation in this organism. This suggests that even if levels are low, it is possible that there are some functional and biologically meaningful 6mA modifications in this organism, even if their paper did not explore this further.

A34. We agree with reviewer to make it clearer we have now changed the sentence to “Although two species in other EDF clades (Rhizoclostridium globosum and Catenaria anguillulae) also have low 6mA abundance, they have either low or no ApT motif methylation”

c) Lines 129-143: This is the same pattern observed in Mondo et al., 2017, where 6mA was most often observed at conserved genes in EDF (for example supplementary table 2).

A35. The reviewer is correct. Added “Thus, 6mA methylation acts on a core gene set rather similar to other EDF than isolate-specific genes.” to improve accuracy.

d) Lines 28-29: Heterogeneity in methylation ratios are observed in other EDF, for example Lobosporangium transversale (Mondo et al., 2017; supplementary figure 1d), although not to the level observed in AMF. Again, while not discussed in their paper, it is important to mention this organism here to provide appropriate context, especially since Lobosporangium (Mortierellomycotina) is also a member of the Mucoromycota (and some phylogenies place Mortierellomycotina as sister to Glomeromycotina).

A36. We agree with the reviewer. We have now added a sentence to include the suggestion in the revised version of the manuscript. Specifically, we added “Some 6mA heterogeneity has been reported in one member of the EDF (Lobosporangium transversal; Mortierellomycotina), but not to the amount observed in R. irregualris”

2) Something seems wrong with the phylogeny in figure 1. The topology is incongruous with most fungal phylogenies produced to date. In Figure 1, the Chytridiomycota and Blastocladiomycota appear as sister to the Dikarya, whereas in most phylogenies the Mucoromycota are sister to Dikarya (for example, see review by James et al., 2020: <https://www.annualreviews.org/doi/full/10.1146/annurev-micro-022020-051835>).

A37. We agree with the reviewer and we have now changed the figure and corrected the phylogeny

3) I cannot find any methods describing how 5mC was detected, except for in Figure 1b where 5mC abundance is mentioned to come from LC-MS data. This needs to be added to methods.

A38. Reviewer is correct, we have now added information pertaining to the detection of 5mC using LC-MS. It was our mistake because during various edits of the manuscript, we removed this information by mistake. (Also see answer A5 to reviewer 1 above).

4) Much of the supplementary material is devoted to structural variant analysis but this is barely mentioned in the main text (only in the 1st section of results where it is merged with description of genome coverage and annotation). I would recommend either removing SV analysis from the manuscript or significantly expanding on it in the main text and explaining how this contributes to our understanding of Glomeromycotina biology.

A39. The current focus of our manuscript is the R. irregularis 6mA methylome. While there is good genome sequence information already generated using Illumina sequencing for some of these fungi this isn't very useful for studying SV. So, before our study, there wasn't much information about SV in these fungi. A fundamental aspect of our study is that we chose to look not just at 6mA methylation in this species but to compare it among isolates, between 2 clones (i.e. as genetically similar as possible; C2 versus C5) and in a dikaryon that has a population of nuclei of two distinct genotypes (isolate C3). This would not have been possible without first looking at SV within and among isolates. So, we do rely on this data and so we think that it is necessary for the reader to be able to access the full information. The SV information helps us to say that C2 and C5 really appear to be clones. In the case of the dikaryon C3, Illumina data shows it is a dikaryon as we can clearly see many bi-allelic loci. However, looking at long-range sequences and SV allows us to say with more certainty whether different sequences were on different nuclei or due to repeats within the same nucleus – something difficult to resolve with the Illumina data. This information is needed to then do the comparisons of the methylome in these isolates but we put it into the supplementary information not to detract from the main story. But we think readers need to have access to the information.

5) Lines 132-133 and 334: I am concerned that the threshold of a single modification is too low to consider a gene as methylated. I wonder if such low abundance is biologically meaningful, especially in AT rich organisms, like R. irregularis, where there is a higher chance of detecting methylated AT dinucleotides by chance. I think 6mA clusters should first be determined before calling genes that are methylated. Alternatively, perhaps genes can be called as 'methylated' if more than a certain minimum # of methylated ApTs (determined based on your data) are present at promoters.

A40. See the detailed answer A20 as this was raised by reviewer 1 as well.

6) The title of this manuscript is too long and unclear. It needs to convey a specific, succinct message. I also suggest removing language like 'reveals their unique place at the crossroads of the fungal tree of life' for reasons stated already.

A41. We have changed the title and made it much simpler. The required maximum title length for the journal is 15 words.

7) Lines 137-143: This is a very interesting part of the paper and could have important implications for the mycorrhizal community. I recommend expanding on the genes that are differentially methylated between genetically identical isolates – what genes are they, and can any explain the differences these isolates have on observed crop growth? Same with differential methylation between nuclei in the dikaryon. I see details on these in the supplementary text – they should be moved to the main text I think.

A42. We understand that differentially methylated genes in clonal isolates C2 and C5 is very important information. However due to lack of functional gene annotation it is difficult to explain/speculate on a per gene basis. Also, lack of a functional annotation does not help further in identification of significant gene ontology enrichment terms. In the case of the dikaryon isolate C3, the analysis was to show that indeed there is differential methylation between the two nucleotypes within the same isolate. We agree that further in-depth research is required with possibly phased genomes of these dikaryon nucleotypes and impact of differential methylation on clonal isolates to identify single gene level functional implications of these results. Actually, the role of methylation in the fungus and its effects on the symbiosis with plants is the current focus of on-going experiments in the group.

Minor comments:

1) Line 140-141: Language needs to be carefully chosen here - 6mA was shown to be associated with actively expressed genes in fungi previously, but not a pre-requisite for gene expression, since unmethylated genes can also be expressed at high levels, as you see both here and in Mondo et al., 2017.

A43. We have changed the sentence to make it clearer "If 6mA regulates gene expression then this could at least partially explain the extremely large differences in symbiotic effects of clones C2 and C5 on crop growth, as well as among clonal siblings"

2) Lines 20-21: "450 million years ago the EDF sub-phylum Arbuscular mycorrhizal fungi (AMF; Glomeromycotina)" Reword, this is confusing. Should be something like: "450 million years ago the Glomeromycotina (Arbuscular mycorrhizal fungi; AMF), a sub-phylum within the Mucoromycota"

A44. Implemented

3) Line 40-41: 1bya refers to the origin of fungi, and needs references.

A45. Implemented

4) Line 45: What does ‘assuring active transcription’ refer to in the context of 6mA? Again, like minor comment 1, language around 6mA’s role in gene expression needs to be chosen carefully.

A46. The reviewer is correct. We have changed this sentence anyway as the abstract had to be greatly reduced in length anyway to conform to the journal's formatting requirements.

5) I noticed several places in the text where the authors were comparing AMF to 'EDF', yet AMF are clearly part of this group. They should compare AMF to 'other EDF'.

A47. Implemented at multiple places in the manuscript.

6) Figure S2 – strand is hard to see in the plot, consider either merging strands together since they show the same distribution, or using a different way to visualize them.

A48. We attached a high-resolution version of the figure so that the reviewers can read the figure. But the figure resolution is lost when inserting it into the Word document required for the journal for reviewing purposes. We hope the reviewers can see the attached high-resolution version.

7) Lines 274-277: this section is unclear and I am having a hard time figuring out what the authors did, how genes were identified for tree building, and how many genes were used.

A49. We are sorry that the section was unclear. We have now changed the manuscript in the methods and supplementary information sections to make this clearer. We identified single copy genes using OrthoFinder and used 6941 genes for this analysis. Specific details can be found in the methods section and supplementary information, Figure S1.

Reviewer #3 (Remarks to the Author):

Arbuscular mycorrhizal (AM) fungi are the ubiquitous symbiotic fungi of plants and promote global phosphate cycling in terrestrial ecosystems. Despite their biological importance, the genomic basis for gene expression regulation has hardly been clarified, because they are coenocytic and unculturable fungi.

The authors found the following points by using the Pacific Biosciences RSII platform with SMRT cell technology.

- AMF are systematically classified as EDF, but the abundance of 6mA is as low as 0.1 to 0.2%, which is different from other EDFs.
- On the contrary, 5mC, which is present only low in other EDFs, is detected frequently (~30%).
- The feature that 5mC is high is rather close to the feature of Dikarya.
- However, although the abundance of 6mA is low, they show a heterogenous pattern among isolates and may be involved in epigenetic heterogeneity and population-level adaptation.
- 6mA are localized in the core fungal genes.
- There is a symmetric pattern at the location of 6mA, which is similar to EDF and is likely to be maintained coupled to DNA replication.
- Enrichment from GO analysis to fungal hydrolase activity and transporter activity can be

seen.

-This is positioned as an important study as an analysis of AMF that was not included in the previous 6mA analysis of EDF.

-It is an important achievement to clarify the characteristics of low 6mA abundance and high 5mC abundance, which is different from general EDF.

-It is interesting to see the difference in the pattern of 6mA among isolates from the viewpoint of causing non-uniform effects on plant growth.

I think these results are highly appreciated, as providing the basis for genome-wide gene expression of AMF.

To strengthen the message of this paper, it is desirable to perform the following analyses.

A50. We thank the reviewer for the comments. We have now performed more analysis and answered the reviewers' questions.

A 6mA difference among isolates has been detected. It seems that there are differences in the relevant methylase genes among isolates. It is advisable to identify the methylase genes from their genomes and show the differences among isolates.

A51. We thank the reviewer for this comment. It was a comment also brought up by some of the other reviewers and led us to some interesting analysis of methyltransferases that we think add a lot to the revised version of the manuscript. We have analysed all the MT-A70 family of genes among isolates with references in Figure S7 and Figure 4 in the main manuscript. We didn't find any difference among isolates for this gene, which is known to maintain symmetric ApT methylation. The sequences among isolates were also almost identical and the catalytic motif (DPPW) was conserved. Therefore, it is more likely that the methylated gene differences among isolates are due to other regulatory mechanisms than MT-A70. However, we cannot clarify the exact mechanism for this because of the general lack of knowledge for fungal genome methylation maintenance. See also A10.

AMF show the presence of high 5mC (Fig. 2b), but it would be even better if there were analytical data comparing it with Dikarya to see where they are located in the genome.

*A52. The current focus of our manuscript is 6mA methylation. We agree with the reviewer that the genome wide organization of 5mC is of very high potential interest and it is something we have now started to work on. However, it requires significant work due to the lifecycle of the fungi and should be appropriately dealt in a separate manuscript. Noted by reviewer we did find clear differences in the levels of 5mC and 6mA in *R. irregularis*.*

Correlation analysis between expression level of genes of interest of isolates by qRT-PCR and 6mA pattern.

A53. We have discussed the association between 6mA patterns and gene transcription in detail above. The analysis we have done shows an overall correlation between the presence of 6mA marks in a gene and their transcription. In our analyses, we have done this for thousands of genes and the result is very consistent across different isolates of the fungus, which would not be expected if the correlation was a false one. To get a similar correlation using qPCR would require doing this for a very large number of genes which is not very

practical and just doing this with a handful of genes probably wouldn't be very meaningful and would still be based on correlation. Actually, a better thing to do would be to grow the fungus in different environments and then hopefully detect differential 6mA patterns (as suggested by reviewer 1). It would then be a case of correlating patterns of gene transcription with presence of absence or different 6mA levels induced experimentally. This is quite straightforward for 5mC (which was not the focus of our study) but very difficult with 6mA, as there is no enzyme-based detection system for 6mA detection at specific sites and would require full genome sequencing and detection of 6mA marks every time for each isolate in each environmental condition (either again using Pac Bio sequencing or using Oxford Nanopore sequencing). This would be prohibitively expensive.

Finally, I think the following subheading does not have sufficient evidence.

"6mA methylation in AMF has a functional role in gene regulation similar to EDF"

A54. The reviewer is correct. We changed the subheading for more clarity "6mA methylation in AMF has a likely functional role in gene regulation similar to other EDF. But in addition, we have changed the text in a number of places in the manuscript to reflect that this is an observed association between 6mA methylation and gene expression and not that we have shown that specific 6mA marks have a direct effect on gene regulation.

Reviewer #4 (Remarks to the Author):

The article described six arbuscular mycorrhizal fungi (AMF) genome assemblies and the epigenome landscapes. In this study, authors found that unlike the other early-diverging fungi (EDF), AMF was more like Dikarya and other eukaryotes, which exhibited relatively low content of 6-methyldeoxyadenine (6mA) but high 5-methylcytosine (5mC) content. The author focused on analyzing the distribution of 6mA in the genome and its relationship with gene expression. They found that the genes containing 6mA are related to the character of arbuscular mycorrhizal fungi, such as phosphate metabolism and DNA methylation. These results were novel. However, after reading the whole story, I am confused by some results.

We hope that our answers below will clarify the confusion.

A55. We thank the reviewer for the suggestion, as well as the appreciation of our study. We answer the concerns of the reviewer below.

First of all, I think the main contribution of this work is providing six high-quality fungal genomes. I suggest that the authors compare the six de novo assembled genomes with other published fungal genomes, such as the orthologous genes or homologous genes, or the PAV or CNV of AMF genomes to the other EDF genomes. I believe that the biggest difference between AMF and other EDFs should be genetic differences. And, epigenetic differences may help increase the adaptability of AMF to different environments.

A56. The model species of the Glomeromycotina, R. irregularis has been sequenced quite a few times using Illumina sequencing and a between isolate comparison as well as a comparison to other fungi has been made have already been published. This would not be novel and is really not the focus of this study. Two breakthrough papers described the methylome of the fungal kingdom but unfortunately did not include the Glomeromycotina

which are an extremely important group of fungi on which a very large amount of research is conducted. There is a clear lack of information on the methylation in the Glomeromycotina. Here we document the interesting features of the methylome in these important fungi.

Second, the authors emphasize that the novelty finding of this study is the relatively low content of 6-methyldeoxyadenine (6mA) but high 5-methylcytosine (5mC) content in AMF genomes. For this result, I think it will better to use several representative EDF and Dikarya geome's methylation contents to highlight the methylation pattern of AMF.

A57. Mondo et al (2017) showed many representative EDF and Dikarya methylomes but not AMF. R. irregularis is a model fungus within AMF and an important member of mycorrhizae phylogeny that has both implications in basic biological understanding of this group of fungi as well in sustainable agriculture. We focused on 6mA because our long-read sequencing allowed the detection of 6mA methylation which we found putatively regulates important genes that have been previously implicated in symbiosis with plants. Furthermore, developing research into 5mC is an important avenue we provide for future research.

In addition, how to detect and calculate 5mC in this research? Since the 5-mC content in the AMF genome is high, why don't the authors pay more attention to 5mC?

A58. As explained in an earlier comment to one of the other reviewers, the 5mC result is very interesting. In our study (which was actually started a long time ago) we intended to also record 5mC marks in the genome using the long-range sequencing results. However, between starting the project and analysing the results, some groups published work indicating that detecting 5mC marks in Pac Bio sequencing data is not fully reliable. The work on the 5mC distribution in AMF is currently a focus of study in our lab but even if we had these results, we think they would probably be too much to add into this manuscript.

Thirdly, how the association between DNA methylation and gene expression were concluded in this study? I notice in the last two or three paragraphs, the authors described that genes contained 6mA were under the regulation of methylation of 6mA. Generally, if we want to say DNA methylation regulates gene expression, we need to get the methylation mutant or use different experimental conditions to prove that relationship. So, I think it is not stringent to conclude that genes were regulated by 6-mA, only base on the fact that genes contain 6mA methylation.

A59. We agree with reviewers, we have now chosen careful wording to avoid directly implicating regulation i.e. "6mA methylation in AMF has a likely functional role in gene regulation similar to other EDF". What we earlier meant here was to suggest that this methylation might have an important role in gene regulation and these genes were affected by methylation. However, to be clear, it is not just the fact that these genes contain methylated adenine, but that they occur in non-random positions at ApT motifs, which are known in some other eukaryotes to have a role in gene regulation. Finally, we have also observed that indeed there is an association between the presence of methylated adenine in gene bodies at ApT motifs and gene expression.

In the last, it is an interesting result that genes contained 6mA were related to DNA methylation. Can authors provide more detailed information on this result, for example, how about the similarity of these genes to well-known methyltransferases in animal and plants?

A60. We have now analysed and added reviewer's suggestion in main manuscript and into supplementary text further for complete explanation. Also see A10.

Reviewers' comments:

Reviewer #2 (Remarks to the Author):

I think this manuscript is much improved over the original version, and has addressed many of my concerns. Primarily, the concern regarding the previous narrative that Glomeromycotina 6mA patterns were unique and represent a transitional state between EDF and Dikarya. I think the updated version reads better and the statements made are more durable. However, the reported global 6mA patterns still don't appear very different from other EDF to me and may quickly lose their 'unique' status as methylomes of other EDF are analyzed. Nonetheless, describing 6mA patterns in AMF and the observation that symbiosis genes appear to retain methylation despite the low %6mA overall are particularly important, especially to the mycorrhizal community.

Regarding global methylation patterns, what is surprising are the authors 5mC results. Although this is stated to be like the Dikarya, this is unlike any eukaryote I am aware of due to the extremely high levels observed. The reported % cytosines methylated is also out of bounds compared to Bewick et al., 2019, who report 5mC abundance in 40 fungi dispersed across the kingdom. It is unfortunate that there is not more exploration of 5mC here, but a logical next step for future study is to acquire single-base resolution data on 5mC to characterize where these modifications are found. Based on results in other fungi, we expect repetitive content to be enriched in 5mC (and there are a lot of repeats in AMF!).

I only have a few comments remaining to help improve manuscript quality:

Major comment:

I don't see figure captions for the figures on the last 2 pages of the merged document, did I miss them?

Minor comments:

- 1) There are various typos throughout the manuscript, particularly in lineage names.
- 2) Figure S2 still looks fuzzy to me.
- 3) Regarding the window considered as part of 'gene body': I am not sure how meaningful 6mA present 500bp downstream of genes is. Given what we know of 6mA in EDF, I would suspect marks here are related to nearby promoters of downstream genes. In other words, the 6mA mods you see 500bp downstream of genes (~1.2% of total 6mA according to fig 3b), in how many cases could you attribute their presence to being within the promoter region of a downstream gene? I would recommend adding a sentence to methods or results about this.

In response to your one of your rebuttal comments:

- 1) 5mC does not appear to be 'almost undetectable' in *Linderina pennispora* from Mondo et al., 2017. While low, it is comparable in abundance to *K. imperatae* (Basidiomycota), shows a clear CpG motif and enrichment at repetitive regions of the genome (Fig S8). The observation of 6mA and 5mC in *Linderina* shows that this can happen in EDF and so the observation of both in AMF is not particularly unique. What is unique is the extremely high levels of 5mC detected in AMF so this aspect should be emphasized.

Reviewer #3 (Remarks to the Author):

I understand the authors' response to the comments.
I have no further comments in particular.

Reviewer #4 (Remarks to the Author):

I am glad to see the manuscript has improved a lot and considered all reviewers' suggestions and comments. The theme is clearer that *Rhizophagus irregularis* exhibited a conserved low 6 mA modification genome-wide, which are probably associated with gene expression regulation. From the answers to all reviewers' comments, it is obvious authors want to emphasize the importance of this finding. However, I think there is still some evidence loss (or not clear) in the current manuscript.

Revealed by the comparison of SV number among all sequenced samples (Figure S3), there were obviously different amounts of SV detected in C2 and C5, however, authors emphasize they were "indistinguishable in SNP level". It is important to clarify whether they are genetically identical or not. Because all the epigenetic variations we want to study are logically based on comparable species or materials without genetic variations or mutations. So, this is why I ask how the genome assembly in my last revision comments. But authors only emphasize this was not the important point in this study, which I can't agree with.

As for the 6mA methylation level comparisons in Figure 2a, only one thing can be concluded that C2, C3 and C5 have similar low methylation levels, but not as authors say that "R. irregularis was much lower than most of the rest of this ancient fungal group...". Here, I suggest adding the published data in this figure to directly compare how the 6mA methylation level of other EDF fungal. Besides, the comparison of 5mC between R. irregularis and other types of EDF and Dikara fungal, it will be better to choose represented species to do the LC-MS for 5mC, too.

It seems not only me but also other reviewers were very interested in 5mC methylation in R. irregularis. As for me, the question is whether the R. irregularis contain DNMTs (DNA methyltransferases, for 5mC), which are very well studied in plants and animals.

In the end, I just realize that Figure 1 occurred in the Introduction section, and perhaps based on the cited literatures, but not the de novo assembled genomes in this study. I am not sure it is appropriate to use it as one of the important results.

Reviewer #5 (Remarks to the Author):

This study describes the DNA methylation landscape of the model arbuscular mycorrhizal species *R. irregularis*. The authors combined long-read sequencing and mass spectrometry to decipher the role of 6mA and 5mC in gene regulation and genome architecture in an AMF genome. They demonstrate that unlike other early-diverging fungi, *R. irregularis* present low levels of 6mA and high levels of 5mC. Even with reduced levels, 6mA is retained in specific genes that seem to play an important role in symbiosis.

Overall the authors have addressed the concerns of the reviewers. They have provided sufficient revisions and supplemental analyses to address all reviewers comments. I do not have additional concerns regarding this manuscript.

I just have one minor comment: L356: "Tools analysing polymerase kinetics" Can the authors cite what tools were used? The reference send to Pacbio tools but it would be nice to have the tools names mentioned in the main text even though details are present in the supplemental data.

We greatly appreciate the comments from all the reviewers. We have carefully considered them and given our responses below. Reviewers' comments in black, our replies in red.

Reviewer #2 (Remarks to the Author):

I think this manuscript is much improved over the original version, and has addressed many of my concerns. Primarily, the concern regarding the previous narrative that Glomeromycotina 6mA patterns were unique and represent a transitional state between EDF and Dikarya. I think the updated version reads better and the statements made are more durable. However, the reported global 6mA patterns still don't appear very different from other EDF to me and may quickly lose their 'unique' status as methylomes of other EDF are analyzed. Nonetheless, describing 6mA patterns in AMF and the observation that symbiosis genes appear to retain methylation despite the low %6mA overall are particularly important, especially to the mycorrhizal community.

Regarding global methylation patterns, what is surprising are the authors 5mC results. Although this is stated to be like the Dikarya, this is unlike any eukaryote I am aware of due to the extremely high levels observed. The reported % cytosines methylated is also out of bounds compared to Bewick et al., 2019, who report 5mC abundance in 40 fungi dispersed across the kingdom. It is unfortunate that there is not more exploration of 5mC here, but a logical next step for future study is to acquire single-base resolution data on 5mC to characterize where these modifications are found. Based on results in other fungi, we expect repetitive content to be enriched in 5mC (and there are a lot of repeats in AMF!).

We greatly appreciate the reviewer's comments and indeed ongoing studies in our group are characterising 5mC patterns in different environments. We have slightly changed the text because of this comment. The reviewer is right that actually 5mC patterns in AMF are actually not like other Dikarya. When we said they were like other Dikarya, we meant that both AMF and Dikarya show high % of 5mC. However, it's true that we report figures even higher than those reported in the Dikarya and so we have highlighted this novelty and made it clearer in the text. Actually, we think this small change improves the manuscript greatly.

I only have a few comments remaining to help improve manuscript quality:

Major comment:

I don't see figure captions for the figures on the last 2 pages of the merged document, did I miss them?

Yes, the reviewer is correct. In the 1st submission, the reviewer questioned the resolution of 2 figures. The original files for these figures are indeed high resolution. However, when the figures are inserted into the manuscript file (which is just for review purposes) resolution in the figures is lost. Therefore, as explained in the previous submission in the "response to reviewers" document, we submitted 2 higher quality versions of two figures to be viewed independently from the merged pdf, if necessary, to allow the reviewer to check details.

There were Figure 5 and Figure S2. They automatically got merged with the pdf by the online submission system but were meant to be viewed separately. This is not a problem. If the manuscript will be accepted for publication, it will not appear as the merged file as Communications Biology require the original figure files for final production and do not use the embedded figures of low resolution that were incorporated in the document for review.

Please note that in the 1st revision, the legends for those two figures are indeed in the manuscript file (for figure 5) and in the supplementary file for figure S2.

We tried to explain this in the response to reviewers in the 1st revision, but probably we were not clear enough.

Minor comments:

1) There are various typos throughout the manuscript, particularly in lineage names.

Thank you for pointing this out. We have corrected these, where we could see them.

2) Figure S2 still looks fuzzy to me.

As explained above, and in the previous “response to reviewers”, we submitted the original figure file for Figure S2 so that the high-resolution version could be opened and checked. Unfortunately, the submission system has merged this into the merged pdf and the reviewer needs to look at the original file to see the true resolution of this figure. This should not be a problem for final production, if accepted.

3) Regarding the window considered as part of 'gene body': I am not sure how meaningful 6mA present 500bp downstream of genes is. Given what we know of 6mA in EDF, I would suspect marks here are related to nearby promoters of downstream genes. In other words, the 6mA mods you see 500bp downstream of genes (~1.2% of total 6mA according to fig 3b), in how many cases could you attribute their presence to being within the promoter region of a downstream gene? I would recommend adding a sentence to methods or results about this.

We did not find the intersection 6mA methylation marks within promoters of other downstream genes within 500 bp in our results. However, it is difficult to define the exact gene boundaries, including promoter/regulatory regions of neighbouring genes, with the genome information currently available for these fungi. In concordance with the reviewer's suggestion, we added a sentence in the methods of our main manuscript to clarify this point and for accuracy.

However, since 6mA marks in these regions represent a very small percentage of the total, it is, as the reviewer points out, a minor point.

In response to your one of your rebuttal comments:

1) 5mC does not appear to be 'almost undetectable' in *Linderina pennispora* from Mondo et

al., 2017. While low, it is comparable in abundance to *K. imperatae* (Basidiomycota), shows a clear CpG motif and enrichment at repetitive regions of the genome (Fig S8). The observation of 6mA and 5mC in *Linderina* shows that this can happen in EDF and so the observation of both in AMF is not particularly unique. What is unique is the extremely high levels of 5mC detected in AMF so this aspect should be emphasized.

This point doesn't directly concern the manuscript itself and we agree with the reviewer.

As mentioned above, we have emphasized the point about high 5mC levels, as suggested by the reviewer.

Reviewer #3 (Remarks to the Author):

I understand the authors' response to the comments.
I have no further comments in particular.

Thank you.

Reviewer #4 (Remarks to the Author):

I am glad to see the manuscript has improved a lot and considered all reviewers' suggestions and comments. The theme is clearer that *Rhizophagus irregularis* exhibited a conserved low 6 mA modification genome-wide, which are probably associated with gene expression regulation. From the answers to all reviewers' comments, it is obvious authors want to emphasize the importance of this finding. However, I think there is still some evidence loss (or not clear) in the current manuscript.

Revealed by the comparison of SV number among all sequenced samples (Figure S3), there were obviously different amounts of SV detected in C2 and C5, however, authors emphasize they were "indistinguishable in SNP level". It is important to clarify whether they are genetically identical or not. Because all the epigenetic variations we want to study are logically based on comparable species or materials without genetic variations or mutations. So, this is why I ask how the genome assembly in my last revision comments. But authors only emphasize this was not the important point in this study, which I can't agree with.

We understand the reviewer's concerns. However, before explaining in detail, why we claim these two isolates are most likely clones (i.e., genetically the same), it is very important to clarify one point. We use the term "genetically indistinguishable" and this term is chosen very carefully rather than saying genetically identical or saying they are clones of each other. With current sequencing technology, it is never possible to claim that 2 individuals are genetically identical. This is because of errors and problematic regions in the reference genome assembly (genome assemblies are never perfect) and slight errors during sequencing. For this reason, if you sequence the same individual twice, you will find differences between the two individuals when you map them to the reference genome. Ultimately, at some point, which is arbitrary, you have to decide whether you think they are

the same, or whether the differences are great enough or believable enough to consider the two individuals to be genetically different.

We refer to Wyss et al. 2016 for variation at the SNP level. Wyss et al. (2016) showed, using short-read sequencing data, that *R. irregularis* isolates C2 and C5 always had the same exact haplotypes at hundreds of polymorphic sites. Thus, ever since, these two isolates were then considered as “genetically indistinguishable”. There are no obvious SNPs that look believable as differences between these two isolates. Having said that, of course there are a small number of SNPs between these two, but when you go manually into the details, in most cases, they just appear to occur at problematic regions of the assembly. So at the SNP level, there isn’t strong evidence for genetic differences, but you can never be 100% sure.

Here, we used our novel long-read sequencing data to investigate whether larger genetic variation (SV) existed between this pair of isolates. To do this, we first compared all SVs detected from their sequencing data when compared to *R. irregularis* isolate DAOM197198 (the reference isolate). The results of this analysis are shown in Figure S3. Indeed, as the reviewer points out, some variation in the number of SVs detected between the two isolates was observed, suggesting the possibility of real genetic variation. Even so, the numbers (approx. 400) represent a negligible difference given the scale. However, the reference genome used in this first analysis corresponded to a highly-fragmented genome assembly, which most likely represented a source of artifacts.

To more reliably investigate variation between C2 and C5, we performed a second analysis where we used our own C2 *de novo* assembly as a reference genome. Using this as the reference, we detected a much lower number of variants when we mapped the sequencing data from C2 and C5. Still, a few hundred sites appeared variable between the two isolates. We then manually inspected whether these variants were truly isolate-specific or not, by visualizing the read alignments in IGV. Indeed, we found that the vast majority of the variants were either the result of sequencing artefacts or had been miss-called in one of the two isolates because of slight differences in sequencing coverage at those particular sites, but that the reads, in fact, supported the call in both isolates. This is exactly why we gave the example of such a case in Figure S4. Hence, we concluded that no convincing SV existed between the two isolates, despite the slight disparity in SV detection. In addition, even if some real variation existed, it would represent a very small fraction of their genome (~0.1%). For this reason, we continued to consider these two *R. irregularis* isolates “genetically indistinguishable”. So, the reviewer is right that if you just look at Figure S3, you might think there are some small differences, but it’s actually completely normal to see such results (as described above). However, our further analyses allow us to discount the possibility of these being real SVs.

The reviewer also questions these results because we want to know whether methylation differences occurred between genetically identical fungi. The reviewer suggests that if some SV actually exists between these 2 fungi, then methylation in one of the isolates at those SV sites, could account for the differences. Actually, we can directly check whether this is an issue. However, if we consider the 400 SVs that occurred in the dataset between C2 and C5 before nay correction for artefacts, we can check how many of these contain apparent 6mA

methylation. In total, we found that out of 461 SVs only 22 SVs harboured 6mA sites. Out of these only one of these contained a gene. Hence, we can confidently say that there was negligible/no effect of SV on m6A status between two clones C2 and C5.

Summarising the above, while it is not technically possible to claim two individuals are absolutely 100% identical, we cannot detect any variation between these two isolates that can be considered as real structural variation or SNP variation, and in all aspects, should be considered as clones. Even taking the apparent SVs shown in Fig S3 (most of which we know are artefacts) hardly any contain methylation and only 1 occurs in a gene. Thus, any SV differences (if they were real) do not affect our methylation results in any meaningful way.

We have added a couple of sentences to clarify this in the main manuscript as well as further clarification in the Supplementary Information.

As for the 6mA methylation level comparisons in Figure 2a, only one thing can be concluded that C2, C3 and C5 have similar low methylation levels, but not as authors say that “*R. irregularis* was much lower than most of the rest of this ancient fungal group...”. Here, I suggest adding the published data in this figure to directly compare how the 6mA methylation level of other EDF fungal. Besides, the comparison of 5mC between *R. irregularis* and other types of EDF and Dikara fungal, it will be better to choose represented species to do the LC-MS for 5mC, too.

The 6mA levels we observed are clearly much lower than most EDF described in Mondo et al. and we maintain this assertion. Most of the EDF sequenced by Mondo et al. have 6mA levels of 2-3%, which is high. In AMF, we saw 6mA of around 0.2% (thus, an order of magnitude lower). In order to see that, it is necessary to read our Fig 2a and compare it with the EDF in Mondo et al.’s Figure 1 (and also compare the scales on the y-axis). We cannot do as the reviewer says and put the known levels of 6mA methylation from Mondo et al. onto our figure for comparison, even if we would like to. We know the exact 6mA levels we measured but we don’t know then for the Mondo et al. data. All we can do is estimate them visually from Figure 1 of Mondo et al. which wouldn’t be accurate enough. Mondo et al. did not report the actual values.

It seems not only me but also other reviewers were very interested in 5mC methylation in *R. irregularis*. As for me, the question is whether the *R. irregularis* contain DNMTs (DNA methyltransferases, for 5mC), which are very well studied in plants and animals.

We agree that it is interesting to check the 5mC methyltransferase conservation in AMF. It has already been reported that the AMF clade has all five types of fungal conserved 5mC MTases (DNMT1, DNMT2, DNMT5, DIM-2 and RID), while sister clades of the same Mucoromycota phylum have lost DNMT5 and RID (Bewick et al., 2019). Together with the divergent 6mA methyltransferases repertoire reported in our study, the conservation of methyltransferases in *Rhizophagus irregularis* is in line with high degree of 5mC methylation found in this fungus and low 6mA level. We have already addressed this issue on pages 204-

211 if the manuscript. We have slightly expanded this section with an extra sentence in the main ms.

In the end, I just realize that Figure 1 occurred in the Introduction section, and perhaps based on the cited literatures, but not the de novo assembled genomes in this study. I am not sure it is appropriate to use it as one of the important results.

This figure 1 represents the summary of the concept in this manuscript and shows the open questions. It was modified in the 1st revision like that following comments from other reviewers. We don't refer to it in the results section and we still think it's appropriate in the introduction to "set the scene" for the study.

Reviewer #5 (Remarks to the Author):

This study describes the DNA methylation landscape of the model arbuscular mycorrhizal species *R. irregularis*. The authors combined long-read sequencing the mass spectrometry to decipher the role of 6mA and 5mC in gene regulation and genome architecture in an AMF genome. They demonstrate that unlike other early-diverging fungi, *R. irregularis* present low levels of 6mA and high levels of 5mC. Even with reduced levels, 6mA is retained in specific genes that seem to play an important role in symbiosis.

Overall the authors have addressed the concerns of the reviewers. They have provided sufficient revisions and supplemental analyses to address all reviewers' comments. I do not have additional concerns regarding this manuscript.

I just have one minor comment: L356: "Tools analysing polymerase kinetics" Can the authors cite what tools were used?

The reference sends to Pacbio tools but it would be nice to have the tools names mentioned in the main text even though details are present in the supplemental data.

We have added the name of the tools, which is simply called "Pacific Biosciences kineticsTools" to the text of the main ms.

REVIEWERS' COMMENTS:

Reviewer #4 (Remarks to the Author):

The authors have addressed my concerns and I can understand all the responses to the comments. I do not have additional concerns regarding this manuscript.